

# A weekly, near real-time dataset of the probability of large wildfire across western US forests and woodlands

Miranda E. Gray[1], Luke J. Zachmann[1,2], Brett G. Dickson[1,2]

[1] Conservation Science Partners, Inc., Truckee, CA 96161, USA
[2] Lab of Landscape Ecology and Conservation Biology, Northern Arizona University, Flagstaff, AZ, 86011, USA

*Correspondence to*: Miranda E. Gray (miranda@csp-inc.org)

**Abstract.** There is broad consensus that wildfire activity is likely to increase in western US forests and woodlands over the next century. Therefore, spatial predictions of the potential for large wildfires have immediate and growing relevance to near- and long-term research, planning, and management objectives. Fuels, climate, weather, and the landscape all exert controls on wildfire occurrence and spread, but the dynamics of these controls vary from daily to decadal timescales.
Accurate spatial predictions of large wildfires should therefore strive to integrate across these variables and timescales. Here, we describe a high spatial resolution dataset (250-m pixel) of the probability of large wildfire (>405 ha) across all western US forests and woodlands, from 2005 to the present. The dataset is automatically updated on a weekly basis and in near real-time (i.e., a one-week lag) using Google Earth Engine and a 'Continuous Integration' pipeline. Each image in the dataset is the output of a machine-learning algorithm, trained on 10 independent, random samples of historic small and large wildfires,
and represents the predicted probability of an individual pixel burning in a large fire. This novel workflow is able to integrate the short-term dynamics of fuels and weather into weekly predictions, while also integrating longer-term dynamics of fuels, climate, and the landscape. As a near real-time product, the dataset can provide operational fire managers with immediate, on-the-ground information to closely monitor changing potential for large wildfire occurrence and spread. It can also serve as a foundational dataset for longer-term planning and research, such as strategic targeting of fuels management, fire-smart
development at the wildland urban interface, and analysis of trends in wildfire potential over time. Weekly large fire probability GeoTiff products from 2005 through 2017 are archived on Figshare online digital repository with the DOI 10.6084/m9.figshare.5765967 (available at https://doi.org/10.6084/m9.figshare.5765967.v1). Near real-time weekly GeoTiff products and the entire dataset from 2005 on are also continuously uploaded to a Google Cloud Storage bucket at https://console.cloud.google.com/storage/wffr-preds/V1, and available free of charge with a Google account. Near real-time
products and the long-term archive are also available to registered Google Earth Engine (GEE) users as public GEE assets, and can be accessed with the image collection ID 'users/mgray/wffr-preds' within GEE.

## 1 Introduction

Spatially-explicit datasets that predict the occurrence and spread of wildfire have taken many forms to address the complexities of wildfire itself, as well as the management and research objectives that guide data development. One



overarching objective amongst researchers has been to understand the risk posed over the course of an individual fire or fire season, contributing to datasets that emphasize the short-term (i.e., days to months) controls on fire (Brillinger et al., 2003; Martell et al., 1989; Sullivan, 2009a, 2009b). Meanwhile, datasets that predict fire occurrence across longer time frames (i.e., years to decades) and larger spatial scales have contributed to another overarching objective to understand the long-term

characteristics of fire regimes. Given seemingly divergent objectives and data needs, datasets have not yet been developed that integrate across short- and long-term environmental dynamics that collectivity contribute to potential wildfire occurrence. Such datasets would effectively relate individual fires to local and synoptic conditions, resulting in predictions at a fine scale that can be scaled up to coarser spatial and temporal resolutions (Taylor et al., 2013). We sought to fill this knowledge gap by developing a novel dataset of large wildfire probability, at a high spatial resolution (250-m pixel) and

across the western US, such that predictions can meet multiple objectives of local to national research, management, and planning efforts.

A well-developed approach to incorporate the dynamic short-term drivers of wildfire is to simulate the spread of a fire with physics-based models (Finney, 2004; Sullivan, 2009c; Tymstra et al., 2010). By simulating individual fires across time and space, this approach is scaled-up to predict the long-term potential of fire at every point on a landscape (Finney et

al., 2011; Parisien et al., 2005). This approach is input- and computationally-intensive, meaning that it requires detailed specifications of many model inputs and is highly sensitive to misspecification of these parameters (Parisien et al., 2012a; Varner et al., 2009). At a landscape-level, this severely constrains the ability of analysts and planners to update datasets on decision-relevant timescales. For example, predictive datasets need to be updated according to changes in fuel that occur within a fire season and on an inter-annual basis.

Alternative methods to predict fire occurrence over the short term relate empirical fire data to environmental predictors in statistical models (Gray et al., 2014; Preisler et al., 2016; Stavros et al., 2014). Data availability in this case, namely the spatio-temporal alignment of accurate and high-resolution fire, weather, and fuels data, also acts as a constraint on the scale of analysis (Taylor et al., 2013). However, such statistical methods are common in predicting fire occurrence on a macro-scale because they can draw on coarse scale data to overcome this constraint (Krawchuk et al., 2009; Moritz et al.,

2012; Parisien et al., 2012b). Owing to the flexibility of model specification and data inputs, as well as increasingly accurate and high-resolution observational data, statistical models can integrate the dynamic short and long-term controls on fire potential.

Indeed, recent studies have explicitly compared the role of temporal scale in predicting fire occurrence, and have shown that long-term norms and short-term fluctuations in climate and vegetation provide complementary predictive power

(Abatzoglou and Kolden, 2011, 2013; Parisien et al., 2014; Riley et al., 2013). For example, long-term climate exerts an influence on the average flammability of a fuel bed, but short-term weather will moderate its flammability in a site-specific way. Similarly, short-term disturbance events such as previous burns can regulate flammability on inter-annual timescales (Parisien et al., 2014; Parks et al., 2015). It follows that predictive datasets of wildfire potential should strive to integrate across complex, dynamic interactions at short- and long-term timescales. Here, we describe a time-series of the probability



of large fire, updated on a weekly basis and in near real-time (i.e., to the present week) to integrate the short-term controls of fire occurrence, but which also considers the longer-term influences of land use, disturbance, climate, and topography. The complete dataset (2005-present) can also be considered a foundational dataset for understanding long-term, probabilistic exposure of forests and woodlands to large fires.

## 5  2 Methods

### 2.1 Modelling

We modeled the probability of large fire occurrence, which we define as the probability that an area on the landscape will burn in a large (i.e., > 405 ha) fire following an ignition event. While defining a large fire size is somewhat arbitrary, 405 ha is commonly used to distinguish large from small fires in western US forests (e.g., Westerling, 2006), and
fires of this size or larger accounted for approximately 95% of area burned in western forests and woodlands from 1992-2015 (Short, 2017). Additionally, our method focuses on the probability of burning in a large fire irrespective of ignition likelihood or sources, which is appropriate given that ignitions are not a primary driver of inter-annual variability in area burned (Abatzoglou et al., 2016).

We used a random forest (RF) classification algorithm to train predictive models of large fire probability. Random
forest is a machine learning technique that recursively partitions variables to classify an outcome of interest, in this case small or large fire events. Multiple classification trees are fit to bootstrapped samples of the training data, but at each node, only a fraction of randomly selected predictors are available for the binary partitioning. The randomized process of recursive partitioning uncovers hidden structures in the data without over-fitting, and yields strong predictive models (Prasad et al., 2006). This makes RF an ideal method to predict fire occurrence across broad and diverse ecoregions, where high
dimensionality is needed to account for unforeseen interactions between climate, fuels, and the landscape.

We derived predictor variables that describe the land surface and climate over multi-year, long-term time frames. Similarly, we derived predictor variables that describe the land surface and weather over weekly, short-term time frames (Table 1). Specifically, an individual fire event on a given day (i.e., a 'fire event day') was spatially related to long-term predictors derived over a multi-year period and short-term predictors derived over the week before and after fire occurrence.
The integration of predictors in this way resolves the dynamic probability of large fire into long-term drivers of fire, and short-term land surface and ambient conditions directly leading up to and following a fire event.

We drew 10 independent, random samples of small and large fires from 2005 through 2014 and used them to train an ensemble of RF models on the set of predictors. Ensemble modeling provides a means of creating classifiers that are more accurate than the individual classifiers that make them up, while depicting the variance across predictions, which is critical
for risk assessment (Dietterich, 2000; Palmer et al., 2005). Using the 10 trained models and gridded datasets of all predictor variables, we created weekly spatial predictions of the mean and standard deviation of fire probability at 250-m resolution across western US forests and woodlands. Predictor variables that were not in a native 250-m resolution were resampled



using bilinear interpolation. Spatial predictions were created for every week from 2005 through the present. See Sect. 4 below that describes the process by which new predictor data acquisitions are automatically and continuously integrated into near real-time predictions and uploaded to the cloud. Models were trained and spatial predictions created within Google Earth Engine (GEE; Gorelick et al., 2016), which is a cloud-based platform that makes terabyte-scale analysis available on

an extensive catalog of satellite imagery and geospatial datasets.

## 2.2 Dependent Variables

We drew 10, independent random samples ($n \cong 900$) of large fire event days from the MODerate-resolution Imaging Spectroradiometer (MODIS) Burned Area (BA) dataset (Roy et al., 2008), which is a global, monthly 500-m gridded product that contains calendar day-of-burn and quality information. The burned-area detection algorithm determines

persistent changes in time series of vegetation indices and then statistically relates those changes to active fire information, in order to characterize them as either burn- or non-burn-related changes (Roy et al., 2005). For modeling purposes, we assumed a temporal accuracy of reported burn date within one day (Boschetti et al., 2010). Before drawing samples, we only retained burned areas that the algorithm flagged as 'most confidently detected' (Boschetti et al., 2015), and that were within eight days of reported burn date of connected burned areas. This boosted our confidence that all remaining connected burned

areas were part of the same fire (Archibald and Roy, 2009), which we further constrained to be at least 405 ha. We masked burned areas according to forest or woodland land cover types classified in the US National Land Cover Dataset (NLCD, 30-m resolution; Homer et al., 2007) before drawing at most one 500-m sample from each burned area (Figure 1).

We drew random samples of small fires from a well-vetted point database of reported fires in the United States (Short, 2014, 2017) that contains day-of-ignition, and which we also masked by NLCD forest and woodland cover. We did

not draw small samples from the BA dataset because the estimated minimum detectable burn size is approximately 120 ha, which means that smaller fires are grossly underestimated (Giglio et al., 2009; Roy and Boschetti, 2009). For future development of this dataset, and recognizing that fire occurrences are not often reported for many regions globally, there are methods that may be adapted to associate active fire information with small fire events (Randerson et al., 2012). For each Environmental Protection Agency (EPA) level III ecoregion in the western US, and for each month and year of burn, we

paired an equal sized random sample of small fires with our large fire sample, resulting in a balanced, 1:1 training dataset across space and time. We restricted the date of fire occurrence to be between April and October in order to only cover the primary fire season across ecoregions, since early-season fires are more likely to be prescribed burns.

## 2.3 Independent Variables

### 2.3.1 Long-term Land Surface Variables

To characterize long-term live fuel availability and fuel moisture, respectively, we used the Enhanced Vegetation Index from the MODIS MOD13Q1 v006 product (EVI, 250-m resolution; Didan, 2015) and the Normalized Difference Wetness



Index (NDWI, 500-m resolution), derived from the MODIS MCD43A4 v006 product (Schaaf, 2015). Both products are 16-day composites computed from atmospherically corrected, bi-directional daily surface reflectance. MOD13Q1 contains pixel quality information and MCD43A4 contains pixel and band quality information. For both products we only retained pixels with processed, good quality data. We extracted five percentile values (10, 25, 50, 75 and 90%) of EVI and NDWI from

2000 (the year MODIS was deployed) to the approximate date of each fire occurrence. These values provided at least five complete years of observed live fuel availability and moisture before fire occurrence.

To characterize the land surface as modified by humans over the long-term, we included an index of human modification in 2001 and 2011 (Conservation Science Partners Inc., 2016; 30-m resolution). This index quantifies the cumulative degree of modification of natural lands attributable directly to energy, residential and commercial, transportation,

and agricultural development. We also used the residential and commercial development dataset alone to compute the distance to urban development in 2001 and 2011. Urban development in this case was approximated by a 'moderate' value of residential and commercial development, which is roughly equivalent to the 'built up moderate' class in the NLCD, except that it removes exaggerated effects of roads. Lastly, we used the Shuttle Radar Topography Mission digital elevation data (Farr et al., 2007) to characterize topographic variables, namely, elevation, slope, aspect, and terrain roughness

(standard deviation of elevation), each at a 30-m resolution.

### 2.3.2 Long-term Climate Variables

We incorporated predictors computed from monthly climatological normals of temperature and precipitation for the period 1980-2010, as derived from the Parameter-elevation Regressions on Independent Slopes Model (PRISM; 800-m resolution; Daly et al., 1994). We selected seven metrics that summarize long-term averages, inter-annual, and intra-annual

variability in climate (O'Donnell and Ignizio, 2012). These included annual precipitation, minimum, maximum and mean monthly temperature and precipitation, seasonality (i.e., the coefficient of variation) of temperature and precipitation, temperature of the wettest and driest months, and precipitation of the coldest and warmest months.

### 2.3.3 Short-term Land Surface Variables

We characterized short-term live fuel availability and fuel moisture with the single EVI and NDWI observations in

the 16 days prior to fire occurrence, and included the absolute value as well as anomalies from the closest day-of-year in years prior and from the five percentile values. We used the MODIS MOD11A2 Land Surface Temperature eight-day composites (LST, 1-km resolution; NASA LP DAAC, 2015), which represent average values of clear-sky LSTs, to similarly characterize the ground temperature immediately leading up to a fire occurrence. We only retained pixels with processed, good quality data. We included both the absolute value of LST from the eight days prior to fire, as well as the LST

anomalies from the five percentile values and from the closest day-of-year in years prior.



### 2.3.4 Short-term Weather Variables

Standard meteorological variables known to influence the daily fire and fuel environment were taken from the gridMET gridded daily surface meteorological dataset (4-km resolution; Abatzoglou, 2013). We incorporated the total precipitation, mean minimum and maximum temperature, mean minimum and maximum relative humidity, mean wind

speed and direction and the mean Palmer drought severity index (PDSI) for the two weeks surrounding fire occurrence.

Standard weather variables have also been compiled into indices that more directly address the processes by which they effect fires and fuels, including the Energy Release Component (ERC), the Burning Index (BI), and 100- and 1000-hr dead fuel moistures (fm100 and fm1000). These indices are components of the US National Fire Danger Rating System (NFDRS) and are derived from models built on the combustion physics and moisture dynamics of the fuel environment

(Schlobohm and Brain, 2002). The fm100 and fm1000 represent the modeled moisture content of large dead fuels and are functions of the latitude, day-of-year, temperature, relative humidity, and precipitation duration in the previous seven days. ERC is a cumulative fuel moisture index reflecting the contribution of all live and dead fuel moistures on the potential heat release, and is also an input into the BI, which additionally incorporates the potential rate of fire spread. gridMET assumes that the persistent fuel environment includes all size classes of dead fuels, as well as herbaceous and woody live fuels, and

all contribute to the derived values of these indices. We incorporated the mean values of ERC, BI, fm100, and fm1000 in the two weeks surrounding fire occurrence.

### 3 Dataset Evaluation

Using the same method described above to create training samples of the dependent variable, we selected 10 random testing sample 'seeds' of small and large fires ($n \cong 400$ each) that occurred from 2015 - 2016 to independently

evaluate the dataset. We extracted predicted values of large fire occurrence at the time and location of testing samples and used the R package 'OptimalCutpoints' (López-Ratón et al., 2014) to determine an optimal cutoff between zero and one that simultaneously maximized the sensitivity and specificity of predictions. Based on an optimal cutoff of 0.45 distinguishing small (< 0.45) from large (> 0.45) fires, the specificity, sensitivity, and overall accuracy of the dataset was 0.79, and the area under the receiver operating curve (ROC) curve was 0.88 (Figure 2).

We took two steps to visualize model performance. We first mapped the rate of false positives and false negatives for each EPA level III ecoregion (Figure 3). Second, we selected 10 random small and large fires from the testing dataset and plotted predictions at these locations over the testing time interval (Figure 4).

### 4 Continuous Integration

We developed a continuous integration (CI) 'pipeline' to generate new predictions as soon as the dynamic predictors

upon which the model is conditioned become available in GEE. The refresh rate of each predictor varies based on the data sources. For example, gridMET assets are updated approximately every two days, whereas the MODIS products are updated



approximately every eight days (Table 1). The pipeline, which tests for the availability of predictors against the requirements of the model, runs on a schedule — compiling each morning at 4am Pacific Standard Time. If all of the criteria are met, a new prediction is generated and appended to the existing collection. We used GitLab.com because GitLab offers continuous integration (CI) services at no cost. The builds are executed using a custom Docker image, which is a bare-bones Ubuntu

5    image configured with the Google Earth Engine Python API client library and its dependencies.

## 5 Band Descriptions

Band 1 - 'mean' :  Mean probability of large fire across 10 trained models. Values range from 0-1.

Band 2 - 'stdDev' : Standard deviation of the probability of large fire across 10 trained models.

Band 3 - 'modis_QA' : For the MODIS products described above, only good quality pixels were retained for model training,

10   but all pixels were retained when creating spatial predictions. Therefore, final predictions contain a band named 'modis_QA' which indicates if one of the short-term MODIS predictors (i.e., MOD13Q1, MCD43A4, or MOD11A2 immediately preceding the prediction date) had unreliable quality.

  0 = All MODIS pixels were processed and good quality

  = At least one MODIS pixel was not processed or had bad quality

**6 Data Availability and Source Code**

  Weekly large fire probability GeoTiff products from 2005 – 2017 are archived on Figshare online digital repository with the DOI 10.6084/m9.figshare.5765967 (available at https://doi.org/10.6084/m9.figshare.5765967.v1). Near real-time weekly GeoTiff products and the entire dataset from 2005 on are also continuously uploaded to a Google Cloud Storage bucket at https://console.cloud.google.com/storage/wffr-preds/V1, and also available free of charge with a Google account.

Near real-time products and the long-term archive are also available to registered GEE users as public GEE assets, and can be accessed with the Image Collection ID 'users/mgray/wffr-preds' within GEE. All source code is available at a GitLab repository (https://gitlab.com/wffr).

## 7 Conclusions

  The dataset we describe here of weekly predictions of the probability of large forest or woodland fire across the

western US invokes interacting effects over multiple timescales that contribute to a site's dynamic fire potential. By drawing on weather, climate, and land surface dynamics at multiple timescales to predict individual fire occurrence at a high spatial and temporal resolution, this dataset overcomes many shortcomings of existing datasets. The result is highly relevant to research, planning, and management objectives that span the western US, ranging from short-term outlooks to long-term planning.

More strategic planning for fuels management is critically needed to adapt to an inevitable increase in wildfire in the west in the coming decades (Schoennagel et al., 2017). For instance, fuels treatments as currently implemented are



limited in their ability to mitigate broad scale effects of wildfire, because it's relatively rare that treatments actually encounter wildfire (Barnett et al., 2016). Strategically targeting areas for treatment based on large wildfire potential, coupled with estimates of burn severity, will lead to more cost- and ecologically-effective decisions. One tool currently used for this purpose is built off of input and computationally intensive models that constrain the ability to update results on timescales

concurrent with the changing fire environment, and the available datasets describe wildland fire potential as of 2007, 2012, and 2014 (Dillon et al., 2015). The dataset we describe here is automatically updated to match the dynamics of the fuel and fire environment, which can easily change and critically effect fuels management decisions on annual timescales.

Another area where strategic fuels and fire planning is critically needed is at the wildland urban interface (WUI). WUI lands in the western US have expanded dramatically over the past few decades, and roughly 40% of these lands are

predicted to experience moderate to large increases in the probability of wildfire in the next 20 years (Schoennagel et al., 2017). Considering also that a large percentage of potential WUI lands are still undeveloped, strategic planning for both fuels management and infrastructure development can make communities more resilient to wildfire. This dataset can help guide development plans at multiple scales (e.g., city, county, or state), drawing on a rich time series that gives analysts and planners access to the observed trends, means, and extremes of the potential for large wildfire over time.

In contrast to longer-term predictions, near real-time predictions of large fire potential provide operational fire managers with immediate, on the ground information to closely monitor how changing conditions affect active fires, and the likelihood that fire suppression will require outside resources. In the US, near real-time predictions are widely used during the peak fire season (Owen et al., 2012). Available products through the US Predictive Services program (http://psgeodata.fs.fed.us/) and the Wildland Fire Assessment System (www.wfas.net; Preisler et al., 2016), consider fuel

and weather conditions changing on daily to weekly timescales, while ignoring longer-term climate and fuel variability that moderate a site's current fire potential. The dataset described here provides near real-time predictions, while simultaneously accounting for dynamic fuel and landscape compositions that are shaped over a longer term, and thus is a significant improvement over operational products of near real-time fire potential.

As the observational record grows longer to include more temporal variability, we will continue to update and

evaluate this dataset. This will allow for any non-stationary relationships between wildfire, climate, fuels, and the landscape to be integrated into predictions. In future development, forecasted climate, weather, and fuels data may also be integrated into the analysis in order to create predictions of large fire probability into the future.

**8 Author Contributions**

MEG developed the fire models and data products, with critical contributions from both LJZ and BGD. LJZ developed the

continuous integration algorithm. All authors contributed to the paper

**9 Acknowledgements**

This research was supported by the Wilburforce Foundation.



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



Table 1. Spatially explicit climate and land surface predictors of large fire probability, including the data source, spatial resolution, and description of how variables were derived from the source data. Gray shading in the table indicates grouping of predictor variables depending on whether they are derived over a short term (sub-monthly) or a long term (multi-year). The accompanying graphic indicates the approximate time period that these variables are drawn from, relative to fire occurrence.

| Predictor | Source | Resolution | Description |
|---|---|---|---|
| *1. Long-term climate variables* | | | |
| Annual precipitation, Min and max monthly temperature/precipitation, Temperature/precipitation seasonality (CV), Precipitation of the coldest/warmest month, Temperature of the wettest/driest month | PRISM | 800 m | Derived from 1980-2010 monthly normals. |
| *2. Long-term land surface variables* | | | |
| EVI percentiles | MODIS | 250 m | $10^{th}$, $25^{th}$, $50^{th}$, $75^{th}$, and $90^{th}$ percentiles from 2000 to the date of fire occurrence. |
| NDWI percentiles | MODIS | 500 m | |
| Human modification, Distance to urban development | CSP 2016 | 30 m | Index or distance value at 2001 for fires pre 2011, and 2011 for fires post 2011. |
| Elevation, Slope, Aspect, Topographic Roughness | USGS | 30 m | |
| *3. Short-term land surface variables* | | | |
| EVI absolute and anomalies | MODIS | 250 m | Absolute value and percent change from 2000 to the fire year, immediately preceding fire occurrence. |
| NDWI absolute and anomalies | MODIS | 500 m | |
| LST absolute and anomaly | MODIS | 1 km | |
| *4. Short-term weather variables* | | | |
| 100-hr fuel moisture, 1000-hr fuel moisture, Burning index, Precipitation, Temperature, Relative humidity, Specific humidity, Potential evapotranspiration, Solar radiation, Wind speed, Wind direction, PDSI | GridMet | 4 km | Mean values in the two weeks surrounding fire occurrence |



Table 1. Cont'd

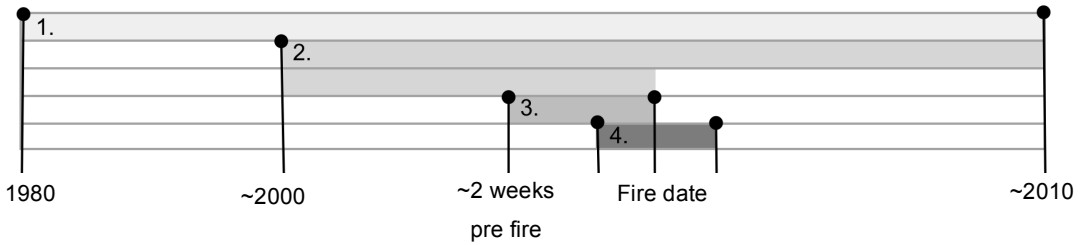



**11 Figure Captions**

Figure 1. Example of how the MODerate-resolution Imaging Spectroradiometer (MODIS) Burned Area (BA) dataset was used to draw 10 independent, random samples of large fire event-days from within large fires. In order to assume pixels were part of the same large fire, connected burned areas had to be within eight days of reported burn date and within three pixels

of each other. MTBS fire perimeters greater than 1,000 acres are included to demonstrate how this approach approximates burned areas within independent burned areas.

Figure 2. Receiver Operating Curve (ROC) for an independent testing dataset of small and large fires that occurred from 2015-2016. Sensitivity and (1-Specificity) values are shown for the point where large fire probability values >0.45 are

classified as a large fire, and values <0.45 are classified as a small fire, since this value was found to simultaneously maximize sensitivity and specificity.

Figure 3. Large fire probability for a week in June and a week in August, 2014. Data layers were taken from the larger dataset of weekly large forest fire probability in the western US from 2005-present.

Figure 4. False positive (FP) and false negative (FN) rates of an independent testing dataset of small and large fires from 2015-2016, mapped across EPA level three ecoregions. No testing data was available for those ecoregions that are not displayed.

Figure 5. Weekly time-series of predicted large fire probability at the location of ten randomly drawn testing samples of small and large fires from 2015-2016. Large and small samples were randomly paired together, and a vertical line is drawn at the date of fire occurrence. A horizontal line is drawn at a cutoff of 0.45, which optimized the sensitivity and specificity of the dataset when distinguishing predicted small (< 0.45) from large (> 0.45) fires.





Figure 1

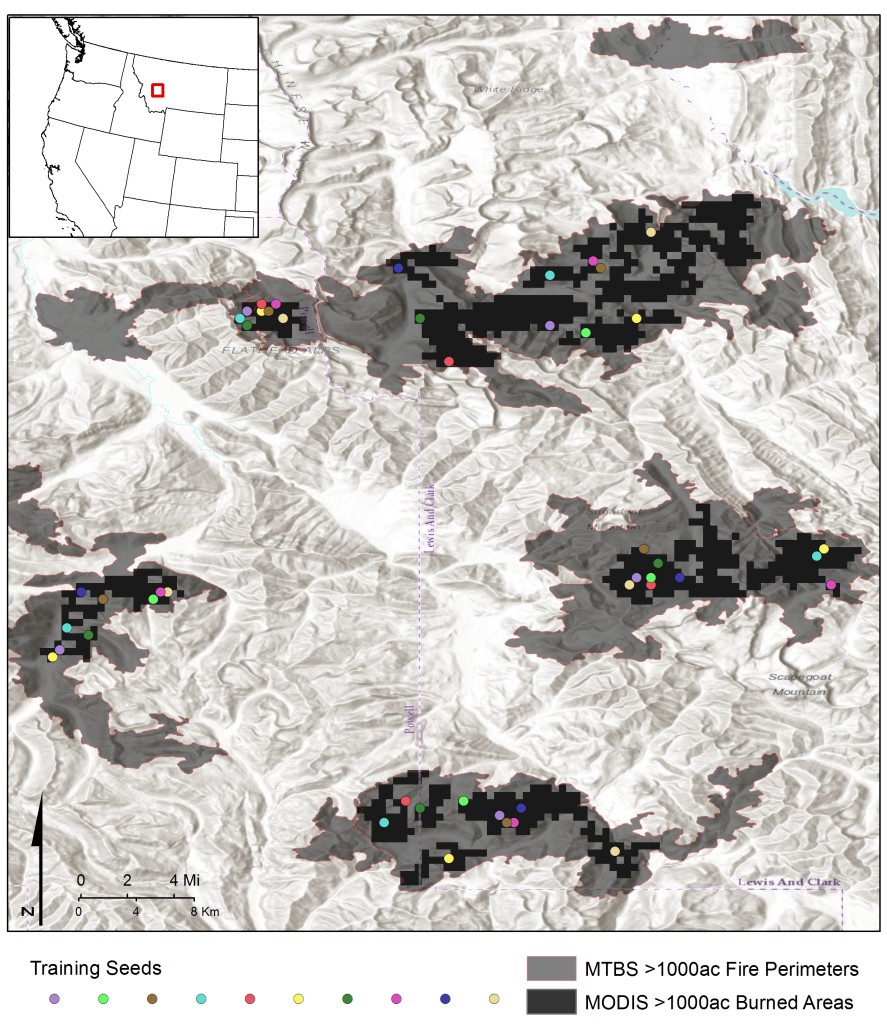

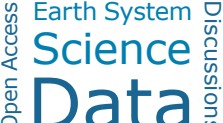



Figure 2

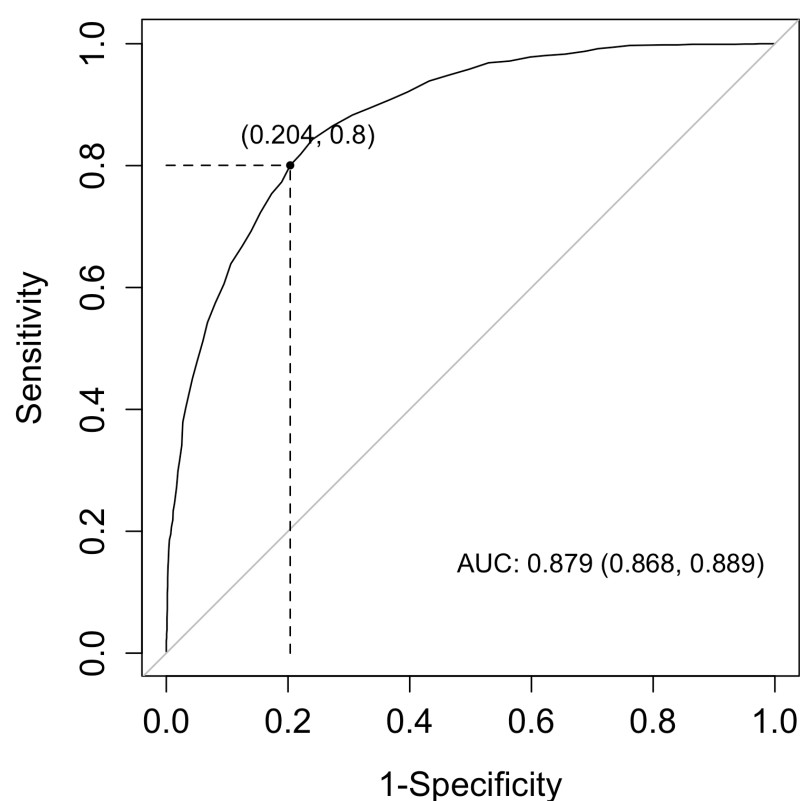

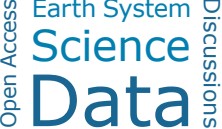



Figure 3

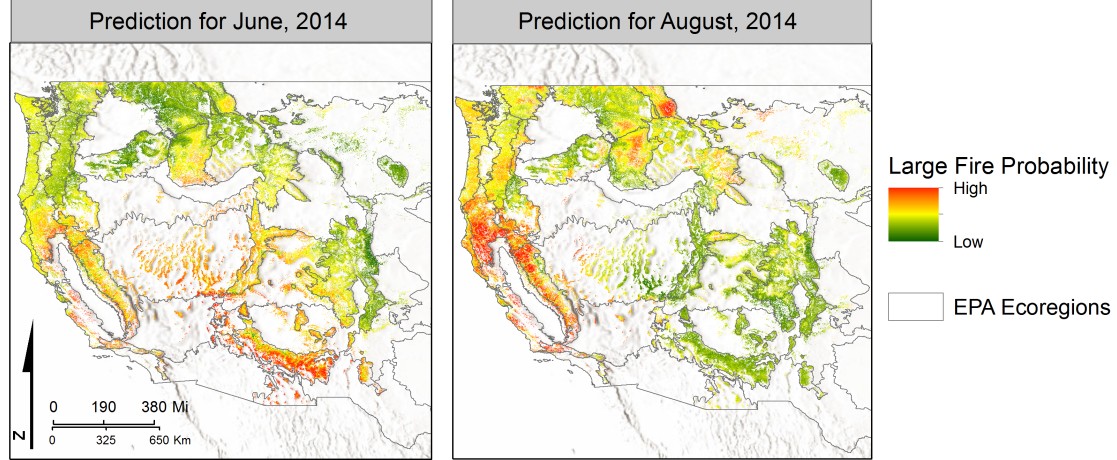



Figure 4

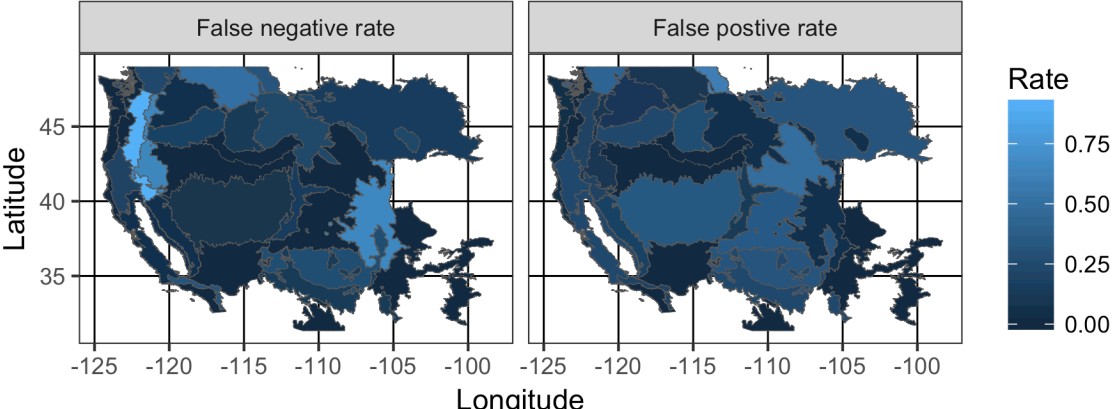



Figure 5