# Peer review of "A weekly, near real-time dataset of the probability of large wildfire across western US forests and woodlands"

_Earth System Science Data, 2017_

## Referee Comment (RC1) · Anonymous Referee #1 · 13 Jan 2018

Review of "A weekly, near real-time dataset of the probability of large wildfire across western US forests and woodlands" by Gray et al.

https://doi.org/10.5194/essd-2017-136

**General Comments:** This is a very ambitious effort to collate numerous disparate databases, develop a model of large fire probability, and create a real-time system for distributing and updating data. I applaud the authors for taking on this challenge. That said, there is substantial room for improvement with respect to the documentation of the modeling effort that should be undertaken before the paper is accepted. I also believe that a more compelling set of figures could accompany the figure to elucidate its utility.

**Specific comments/considerations**
1. The modeling framework is poorly described. The authors talk a lot about the data inputs, but it's not transparent how the model was created (other than RF), whether a single model was used for the entire region, how exactly the 10-different models compare, which variables of those selected were significant predictors.
2. The validation of the data/model is poorly described making it challenging to assess whether the model is "good". This needs to be improved upon for the credibility of the dataset.
3. It is a bit challenging to understand the mismatch in spatial scales that define a large fire (400ha) and the spatial scale of output (250m). Technically, a large fire would consists on a continuous group of at least 64 250m x x250m pixels. The configuration of fuels and topography would appear to be important in determining if a given pixel could be part of a large fire. This would require not only considering the fuel/topography/weather at each individual pixel, but neighboring pixels. I don't know how this is dealt with in the model.

**Technical Corrections/Considerations**
1. Line 25, I am unsure whether it is appropriate to provide links to cloud storage or GEE image collections that may be temporary and not available in a few years time. I think the link to the doi is appropriate, and would suggest keeping the links to the other sources in the main text. This is for the journal to decide though.
2. The first paragraph is a bit rough and could use revision. Specifically you are contrasting the research and operational needs that operate at different spatiotemporal resolutions and highlighting the different data needs. I think the point should be made that while many datasets exist to support such efforts, and effort to synthesize and model large fire risk using an empirical framework at such fine-scales is a gap.
3. Line 9, Page 2, insert "moderate temporal resolution (weekly)" to emphasize that you are not only modeling at high-spatial resolution.
4. Line 16, Page 2, replace "misspecification of these parameters" to "assumed model parameterizations"

5. Line 19, Page 2, I believe that models of fire spread potential are run operationally on fires with FlamMap. It might be best to specifically address the upshot of the proposed modeling platform here over FlamMap (e.g., different goals, spatial extent).
6. Line 29, Page 2, "show that both long-term normals and variability in climate and vegetation…"
7. Line 31, Page 2, It might be useful to specify how flammability is being used here and throughout. It could refer to fuel dryness, but also fuel abundance, or their combination. Long term climate exerts an influence on biomass production and the biogeography of vegetation and hence sets the stage for fuel in addition to average fuel-moisture which limits fire. Also, it might be useful somewhere in the text to specify the timescales of "short-term weather". To some, this is on the matter of hours-to-days in terms of wind-driven fire. In this paper it likely refers to sub monthly timescales related to vegetation dryness.
8. Line 32, Page 2, Rather than say flammability here, I think you really mean biomass or fuel abundance.
9. Line 8, Page 3, Does ignition refer to anything that would potential spread fire into a pixel (ignition, or fire spreading into region)?
10. Line 12, Page 3, It is probably worth caveating somewhere that ignitions are not random, but adhere to specific spatial patterns tied to anthropogenic activity or lightning. It is fine to state that this is not part of the modeling framework.
11. Line 8, Page 4, How are prescribed fires excluded here? MODIS will pick up large prescribed fires. You could use the Short FPA data to exclude any large fires that might have been prescribed. Note that your seasonal window April-October will not eliminate prescribed fires as they occur in many regions in late Sep/Oct. That said, I doubt they are a sizable number given the size thresholds used for large fires.
12. Line 16, Page 4, Is this NLCD data specific to a certain year. Ideally, this could be land cover pre-fire.
13. Line 18, Page 4, Please define size threshold for small fires.
14. Line 25, Page 4, I understand wanting a balanced database. But what consequences do you think there are by selecting small fires in the same month/year from ecoregions where a large fire occurred. There is a lot of spatiotemporal autocorrelation in your primary drivers of temporal variability in your model that could weaken your relationships. Presumably, you'd want random samples of small fires from ecoregions.
15. Line 30, Page 4, I don't complete understand how EVI and NDWI were used here. Long term averages implies averaging over years and months. But in many regions that are snow covered, you would have poor EVI data that is unrepresentative. It would also be useful to provide a basis for using these variables for live fuel moisture .
16. Line 20, Page 5, I think you used 1981-2010 climate normals, as this is the standard 30-year period. I am confused by the variables listed. I believe none of these are inter-annual. Furthermore, if you used monthly temperature and precipitation, what additional value is there in having annual sum/averages, and CV.
17. Line 27, Page 5, See Boer et al. (2017) for LST as a proxy for fire danger.
18. Line 7, Page 6, The NFDRS typically involves a fuel model. What fuel model was used and was is consistent spatially?

19. Line 19, Page 6, How many large fires occurred in total in 2015/16? I think that by "randomly" using 400 large fires, and pulling from a number not much larger than that is a limiting factor (e.g., not very independent).
20. Line 25, Page 8, How do you suspect the model handles non-stationarity? It was built with historical conditions.
21. Table 1: Missing ERC from section 4. Table 1 figure is a bit awkward and might not be needed.
22. Figure 1 caption: Use SI units. Figure 1 itself shows MTBS. Why introduce MTBS here? Also, do you require a different training seed within each large fire? Note that these really won't be independent points due to autocorrelation.
23. Figure 3, I think it is best to just show a single map, but specify the exact date, and perhaps show any large fires that actually occurred.
24. Figure 5, This is a poor figure. I don't understand what the 10 different points refer to, why the 0.45 cutoff is shown, and didn't realize that there was a model for small fires in addition to large fires. It also looks like this includes data through 2017. If so, please update the caption.

References:

Boer, M. M., Nolan, R. H., Resco De Dios, V., Clarke, H., Price, O. F. and Bradstock, R. A. (2017), Changing Weather Extremes Call for Early Warning of Potential for Catastrophic Fire. Earth's Future. doi:10.1002/2017EF000657

---

## Referee Comment (RC2) · Anonymous Referee #2 · 17 Jan 2018

While this manuscript presents a concept that would be valuable (near real-time large fire probability), I believe it oversells its novelty over existing products and I am suspicious that there are methodological flaws (but there is not enough information in the methods section to tell for certain). As the topical editor has already noted, the products are not available at either Figshare or Google Cloud Storage (I did not try the Google Earth Engine). While the authors did locate other near real-time products that deliver similar information (e.g. Preisler et al 2016), they missed other datasets currently commonly used for planning fuel treatments and other management activity (notably Short et al's "Spatial dataset of probabilistic wildfire risk components for the conterminous United States" at https://www.fs.usda.gov/rds/archive/Product/RDS-2016-0034/, which

was calibrated to previous fires using Short's Fire Occurrence Database which also provides part of the underpinnings of this current effort). This dataset and similar efforts using the FSim model (Finney et al 2011) already deliver many of the functions that the current effort proposes it is uniquely positioned to fill, including acting as "a foundational dataset for longer-term planning and research, such as strategic targeting of fuels management" (e.g. Scott et al 2016, Thompson et al 2017), "fire-smart development at the wildland urban interface" (e.g Haas et al 2013), and "analysis of trends in wildfire potential over time" (Finney et al 2011, then Short et al's updated dataset, and also the fire potential datasets the authors reference by Dillon). Given the mature state of this previous work and its prevalence, its inclusion in the current manuscript would seem important. In addition, the authors state that other existing models do not account for long-term fuel and climate variability, but I'm not convinced their model better accounts for these due to a number of reasons: 1) their model uses a number of variables as proxies for fuel and weather that as far as I know have not been demonstrated in the literature to relate to fuel availability and flammability or to meaningfully measure weather's effect on fire, including PDSI (which in fact has been demonstrated to not be related to large fire activity (Riley et al 2013)), EVI, and NDWI, 2) as far as I can tell, their model has a static development layer (the CSP from 2001 and 2011), thus not capturing development trends except coarsely, and 3) it's unclear how their model captures prior burns, which they state are important factors in fire spread (perhaps they mean the EVI to do this, and indeed it might, but citations or analysis are needed to show that the EVI captures differences in the pre- and post-fire landscapes). Other functions that they state their model can accomplish are already delivered during wildfire incidents by a suite of models in the Wildland Fire Decision Support System, including FlamMap, FARSITE, and FSPro – which do in fact function in real time, with runs taking on the order of 15 minutes to 1.5 hours and being delivered the same day to suppression forces. However, the approach taken here is novel and could be a valuable addition to the suite of existing models – if methodological questions can be addressed.

Moving on to model structure, I can't discern from the manuscript what the response variable in random forests is. Is it probability of large fire? Probability of small fire? Wouldn't these two be related and increase together? Is probability that the pixel doesn't burn calculated? Or is it simply predicting binary large versus small fire? Or to predict binary burning by large fire versus not burned? Also, it's unclear how predictor variables were chosen. Did the authors choose a set of variables based on their hunches about what's important? Or was variable importance in random forests used to guide selection? I am concerned about a potential flaw in model design: it appears that the authors model probability of each pixel independently – however, probability of burning is not independent and is affected by contagion from neighboring pixels. Each pixel's burning can't be considered an independent event, since burning is spatially related to its neighbors. To that end, each large fire might properly be regarded as the unit of prediction, not individual pixels. If the authors have accounted for this, methods should be shared in the paper. The manuscript is largely lacking in assessment of goodness-of-fit of the model. Also, is it feasible to predict burn probability without first predicting ignition? Burning is predicated on ignition taking place first, and then on spatial contagion: thus, the burn probability of one pixel wouldn't be independent of the nearby ones. Some discussion of this, demonstrated by out-of-bag error rates and additional goodness-of-fit metrics is needed. Also, better demarcation of prescribed fires versus wildfires is needed. Lastly, how were the outputs validated? By comparison with other existing burn probability or fire regime datasets?

Specific comments follow:

Page 2 Line 4: the sentence that ends "characteristics of fire regimes" would seem to require citations. Please add.

Page 2 Line 7: also see Finney et al 2011, Preisler et al 2016, and Preisler et al 2004, for example, who have already produced this type of work.

Page 2 Line 14: it seems that Finney et al 2011 (FSim paper below) should also be

referenced here. The comment that follows is not relevant to FSim ("it requires detailed specification of many model inputs and is highly sensitive to misspecification of these parameters"), which is calibrated to fire occurrence in Short's FOD.

Page 2 Line 16-18: I am confused by this comment, since as noted above, FlamMap runs in seconds, FARSITE in minutes, and FSPro in 15-90 minutes, meaning that they can and are updated subdaily during active fires.

Page 2 Line 19-20: these lines state that models like Preisler et al 2016 are constrained by availability of accurate high-resolution fire, weather and fuels data – however, later in the manuscript the authors correctly acknowledge that the Preisler et al 2016 model was run daily last year with updated weather data, with outputs available on the WFAS website.

Page 3 Line 24-26: I'm curious how fuel type (grass, brush, timber litter, etc) is accounted for in the model, as fuel type is directly related to fire spread and probability.

Page 4 Line 6: What are "large fire event days"? I'm thinking you mean an individual burned pixel. Please elaborate in this paragraph on how you decided whether burned pixels were part of the same fire – from Figure 1, it looks like you assigned pixels to MTBS fires.

Page 4 Line 23: what is meant by "there are methods that may be adapted to associate active fire information with small fire events"?

Page 4 Line 24: Why was it important to have the same number of small and large fires? There are many many more small fires than large fires in Short's FOD.

Page 4 Line 25-26: There are other (perhaps more effective) ways to remove prescribed burns from your dataset. For example, fire type is an attribute in MTBS. By using April-October fires only, you'd include most Rx burns in northern states like Montana and Idaho, and exclude large southern California fires like the recent Thomas Fire that often take place in December.

Page 4 Line 30: I must confess I'm not familiar with the NDWI, but when I Google it, USGS calls it the "Normalized Difference Water Index" rather than wetness index, and describes it as being used to discern water from non-water. Are you talking about the same index? It seems improbable that there are two MODIS NDWI's with different calculations. . .but perhaps that is the case. Please clarify. In either case, I've not seen the NDWI used in any studies relating it to fire occurrence. So it is a good choice here? The relationship of canopy moisture and flammability is quite complex and not well understood (see for example McAllister et al 2012). Despite the lack of study of the NDWI, it could be a good predictor in your model, but not enough information on variable importance is presented in the current version for me to assess.

Page 5 Line 4-5: If I am understanding correctly, you only used MODIS values from inception up to the date of the fire to assign percentile values. Why not use the whole record? It seems in your current method, the percentile assignments would be sensitive to the date of the fire (so if a fire occurs at an index value of 100 in 2005 and another fire at a value of 100 in 2010 in the same pixel, these could be calculated to be different percentile values since the underlying distribution of values would be different).

Page 5 Line 6-7: What is the index of human modification supposed to signify with regards to burn probability? Why is it included? What was the variable importance score?

Page 5 Line 20: Can you explain more about why the CV of temperature and precipitation is "seasonality"? I don't follow. Similarly, why are temperature of the wettest and driest months and precipitation of the coldest and wettest months included as predictors? Have these been demonstrated to correlate with fire probability? Do they have high variable importance scores?

Page 5 Line 23: I think you are saying that EVI is related to fuel availability. I think you are working only in forested ecosystems, so most times EVI will be correlated with the canopy rather than the understory. However, surface fire propagates in the understory

and crown fires are relatively rare. So is EVI really related to fuel availability? Also, see McAllister et al regarding live fuel moisture and flammability. Also in this paragraph, if you have only five years of data in some cases and you are calculating anomalies, you would have only 5 observations, right? Again, why not use the full MODIS record (or did you)? Also, are the average LSTs for both night and daytime temperatures?

Page 6 Line 5: Why was PDSI included as a variable when it has been demonstrated not to be strongly correlated with large fire activity (e.g. Riley et al 2013)?

Page 6 Line 8: Please state which NFDRS fuel model the ERC was calculated for. I believe Abatzoglou's product is for fuel model G.

Page 6 Line 11: fm1000 represents the previous 42 days (1000 hr/24 = 41.666 days).

Page 6 Lines 18-29: Were small fires assigned to a single pixel? Please explain why only one year of fires was used in evaluation (do you expect these relationships to be stationary from year to year when there is so much annual variability in area burned?). For the rest of this paragraph and the following paragraph I'm quite confused. I don't understand what the response variable in the model is (as stated above). Also, can you briefly define sensitivity and specificity? If the response variable is probability, how do you define a false negative and false positive?

Page 7 Line 10: I would like to see each of the bands illustrated by a figure, otherwise it's quite difficult for a reviewer to visualize and assess the product. How prevalent were pixels with a rating of 1 (at least one MODIS pixel was not processed or had bad quality)?

Page 8 Line 7: again, see other literature including but not limited to Thompson et al 2017 and Scott et al 2016.

Page 8 Line 11: There are other products updated daily that account for changes in weather and fuel moisture, including Preisler et al 2016. Some of the inputs to your model appear to be static, including the CSP (human development layer) and it's not

clear how past disturbances (burns) are included in your model. Perhaps the EVI captures previously burned areas, but I know of no study that documents that. Have you assessed how your model works in recently burned vs. burned areas?

Page 8 Line 20: It's not clear how this model would provide better information to managers during active fires than the suite of models in WFDSS (FlamMap, FARSITE, and FSPro), which output information on predicted fire intensity, fire spread, and burn probability in near real-time. Please clarify.

Figure 1: This figure nicely illustrates how incomplete MODIS data is!! I've noticed this while daily following nearby fires in my area. MODIS often misses surface fires where the canopy is dense or even crown fires where the smoke plume is dense. Is MODIS then a good basis for predicting burned pixels (especially when it can be difficult to eliminate Rx fire)?

Figure 3: Please present actual values rather than "high" or "low". I don't feel I can validate the product without them.

Figure 4: I'm confused here. Why not present at-pixel values? Are these the sum or average of false positives for an ecoregion? I'm also confused as to what these mean: there is always a probability of fire, so what does it mean to have a false negative or false positive if you are predicting probability? Of course, as I said earlier, I'm confused about what the response variable is, so when I understand that perhaps I won't be confused here.

Figure 5: I'm confused here too. Is the white squiggly line the probability of small fire and the black squiggly line the probability of large fire? If so, the y-axis is incorrect. Is the vertical white line the date of a small fire, and the black vertical line the date of a large fire? What does it mean to randomly pair a large and small fire? Should they be related? Why in some cases are the black and white trends similar and in some cases different?

References

Finney, Mark A., Charles McHugh, Isaac Grenfell, and Karin L. Riley. 2011. A simulation of probabilistic wildfire risk components for the continental United States. Stochastic Environmental Research and Risk Assessment 25:973-1000. DOI: 10.1007/s00477-011-0462-z.

Haas, Jessica R., David E. Calkin, and Matthew P. Thompson. 2013. A national approach for integrating wildfire simulation modeling into Wildland Urban Interface risk assessments within the United States. Landscape and Urban Planning 119: 44-53.

McAllister, S., I. Grenfell, A. Hadlow, W.M Jolly, M. Finney, and J. Cohen. 2012. Piloted ignition of live forest fuels. Fire Safety Journal 51, 133-142.

Preisler, Haiganoush K., David R. Brillinger, Robert E. Burgan, and J.W. Benoit. 2004. Probability based models for estimation of wildfire risk. International Journal of Wildland Fire 13: 133-142.

Scott, Joe H., Matthew P. Thompson, and Julie W. Gilbertson-Day. 2016. Examining alternative fuel management strategies and the relative contribution of National Forest System land to wildfire risk to adjacent homes – A pilot assessment on the Sierra National Forest, California, USA. Forest Ecology and Management 362: 29-37.

Thompson, Matthew P., Karin L. Riley, Dan Loeffler, and Jessica Haas. 2017. Modeling fuel treatment leverage: encounter rates, risk reduction, and suppression cost impacts. Forests 8(12), 469; doi:10.3390/f8120469.

---

## Author Comment (AC1) · 12 Apr 2018

R = Reviewer Comment, A = Author Response

Review of "A weekly, near real-time dataset of the probability of large wildfire across western US forests and woodlands" by Gray et al. https://doi.org/10.5194/essd-2017-136

R: This is a very ambitious effort to collate numerous disparate databases, develop a model of large fire probability, and create a real-time system for distributing and updating data. I applaud the authors for taking on this challenge. That said, there is

substantial room for improvement with respect to the documentation of the modeling effort that should be undertaken before the paper is accepted. I also believe that a more compelling set of figures could accompany the figure to elucidate its utility.

Specific comments/considerations 1. The modeling framework is poorly described. The authors talk a lot about the data inputs, but it's not transparent how the model was created (other than RF), whether a single model was used for the entire region, how exactly the 10-different models compare, which variables of those selected were significant predictors.

A: We've added text to make it more transparent how the models were created (Starting pg 4, line 1). We've also added more to the 'Dataset Evaluation' section to address how the 10 different models compared and variable importance. Briefly, using all training data from 2005-2015, we compared models and extracted variable importance. Across the 10 models, overall accuracy was consistently from 0.7 - 0.72 and AUC was consistently from 0.78-0.79. More importantly, we examined the main differences across models in the top 20 important variables. Human modification or distance to urban development, the mean BI, and the lowest percentiles of NDWI were consistently in the top 5 variables. Topographic roughness and mean ERC were consistently in the top 10 variables. Slope, LST, mean 1000-hr FM, the highest percentiles of NDWI, precipitation of the warmest month, and EVI (both short- and long-term) were consistently in the top 20 variables. Other variables showed up inconsistently in the top 20 variables. Please see pg 7 and 8, lines 29-31 and 1-6.

R: The validation of the data/model is poorly described making it challenging to assess whether the model is "good". This needs to be improved upon for the credibility of the dataset.

A: As indicated, we added some of the above to the 'Dataset Evaluation' section of the manuscript, and also clarified the evaluation protocol on an independent testing dataset of 2015-2016 fires. See pg 8 lines 7-19. We also revised Figure 3, which we

believe will add to the credibility of the dataset.

R: It is a bit challenging to understand the mismatch in spatial scales that define a large fire (400ha) and the spatial scale of output (250m). Technically, a large fire would consists on a continuous group of at least 64 250m x x250m pixels. The configuration of fuels and topography would appear to be important in determining if a given pixel could be part of a large fire. This would require not only considering the fuel/topography/weather at each individual pixel, but neighboring pixels. I don't know how this is dealt with in the model.

A: A moving window analysis was used to take the mean values of predictor variables in a circular kernel with a radius of 1135 meters. This results in a window size that is approximately the size of a large fire (i.e., 405 ha), and assumes that this entire area has an influence on whether the focal pixel is likely to burn (e.g., fire may spread from a distant source). Please see pg 5, lines 13-15.

R: Technical Corrections/Considerations

Line 25, I am unsure whether it is appropriate to provide links to cloud storage or GEE image collections that may be temporary and not available in a few years time. I think the link to the doi is appropriate, and would suggest keeping the links to the other sources in the main text. This is for the journal to decide though.

A: Our expectation is that these links will remain up-to-date as long as the doi is current, and so have retained them in the abstract and main text. We will leave it up to the journal/editor to also comment on this issue.

R: The first paragraph is a bit rough and could use revision. Specifically you are contrasting the research and operational needs that operate at different spatiotemporal resolutions and highlighting the different data needs. I think the point should be made that while many datasets exist to support such efforts, and effort to synthesize and model large fire risk using an empirical framework at such fine-scales is a gap.
A: We agree and have revised the first paragraph to make the distinction between research and operational needs clearer.

R: Line 9, Page 2, insert "moderate temporal resolution (weekly)" to emphasize that you are not only modeling at high-spatial resolution.

A: Done. Pg 2, lines 7-8

R: Line 16, Page 2, replace "misspecification of these parameters" to "assumed model parameterizations"

A: Done. Pg 2, line 23

R: Line 19, Page 2, I believe that models of fire spread potential are run operationally on fires with FlamMap. It might be best to specifically address the upshot of the proposed modeling platform here over FlamMap (e.g., different goals, spatial extent).

A: Yes, fire spread potential is run operationally on fires with FlamMap and other programs (e.g., Farsite and FSPro). We've specifically addressed this and any differences with our data product on pg 2, lines 11-19.

R: Line 29, Page 2, "show that both long-term normals and variability in climate and vegetation…"

A: Done. Pg 3, line 4

R: Line 31, Page 2, It might be useful to specify how flammability is being used here and throughout. It could refer to fuel dryness, but also fuel abundance, or their combination. Long term climate exerts an influence on biomass production and the biogeography of vegetation and hence sets the stage for fuel in addition to average fuel-moisture which limits fire. Also, it might be useful somewhere in the text to specify the timescales of "short-term weather". To some, this is on the matter of hours-to-days in terms of wind-driven fire. In this paper it likely refers to sub monthly timescales related to vegetation dryness.

A: Thank you for catching this. We have clarified our use of the term 'flammabil-ity' and have operationally defined our use of the term 'short-term weather' (weekly timescales), on pg 3, lines 6 and 11.

R: Line 32, Page 2, Rather than say flammability here, I think you really mean biomass or fuel abundance.

A: That is correct. We have clarified on pg 3 line 8

R: Line 8, Page 3, Does ignition refer to anything that would potential spread fire into a pixel (ignition, or fire spreading into region)?

A: Yes, because the response variable included both small-fire ignitions and pixels that were part of a large fire, we meant ignition to include both ignition and fire spreading into a region, and have clarified on pg 3 line 18.

R: Line 12, Page 3, It is probably worth caveating somewhere that ignitions are not ran-dom, but adhere to specific spatial patterns tied to anthropogenic activity or lightning. It is fine to state that this is not part of the modeling framework.

A: We have added this caveat and also a citation on pg 3, lines 22-23.

R: Line 8, Page 4, How are prescribed fires excluded here? MODIS will pick up large prescribed fires. You could use the Short FPA data to exclude any large fires that might have been prescribed. Note that your seasonal window April-October will not eliminate prescribed fires as they occur in many regions in late Sep/Oct. That said, I doubt they are a sizable number given the size thresholds used for large fires.

A: Large prescribed fires were excluded from the data used in our analyses by only se-lecting large fire samples from the MODIS BA dataset that were within MTBS wildfires. Please see pg 4, lines 25-27.

R: Line 16, Page 4, Is this NLCD data specific to a certain year. Ideally, this could be land cover pre-fire.

A: We used the NLCD from 2001. Given our focus on fires from 2005-present, it reflects the best (and most contemporaneous) pre-fire land cover. We clarified this on pg 4, lines 30-31.

R: 13. Line 18, Page 4, Please define size threshold for small fires.

A: Done. Pg 4, line 2

R: Line 25, Page 4, I understand wanting a balanced database. But what consequences do you think there are by selecting small fires in the same month/year from ecoregions where a large fire occurred. There is a lot of spatiotemporal autocorrelation in your primary drivers of temporal variability in your model that could weaken your relationships. Presumably, you'd want random samples of small fires from ecoregions.

A: We agree that the way we had set up the sampling was overly restrictive and may have weakened some of the real temporal signal. As suggested, we only took equal samples from within ecoregions to maintain the spatial balance.

R: Line 30, Page 4, I don't complete understand how EVI and NDWI were used here. Long term averages implies averaging over years and months. But in many regions that are snow covered, you would have poor EVI data that is unrepresentative. It would also be useful to provide a basis for using these variables for live fuel moisture.

A: We have added text to justify our use of long-term EVI and NDWI, and clarified that both of these variables were masked for snow and ice before taking long-term percentile values. We were incorrect in stating that NDWI can be used directly for live fuel moisture without also accounting for the amount of vegetation, so we've provided a revised basis (with appropriate references) for using the NDWI to get at water content per pixel, as well as live fuel moisture when coupled with EVI. Please see pg 5, lines 18-32.

R: Line 20, Page 5, I think you used 1981-2010 climate normals, as this is the standard 30-year period. I am confused by the variables listed. I believe none of these are

interannual. Furthermore, if you used monthly temperature and precipitation, what additional value is there in having annual sum/averages, and CV.

A: Yes, we used 1981-2010 climate normals and have revised the text. We've also winnowed this list of bioclimatic predictors down to ones that are least correlated and have been used in previous fire-climate studies. Please see pg 6, lines 20-26.

R: Line 27, Page 5, See Boer et al. (2017) for LST as a proxy for fire danger.

A: Yes, and also Nolan et al. 2016 which we've also included in the text on pg 7, lines 5-7.

R: Line 7, Page 6, The NFDRS typically involves a fuel model. What fuel model was used and was is consistent spatially?

A: Fuel model G was assumed in the GRIDMET dataset and was spatially consistent. Please see pg 7, line 20.

R: Line 19, Page 6, How many large fires occurred in total in 2015/16? I think that by "randomly" using 400 large fires, and pulling from a number not much larger than that is a limiting factor (e.g., not very independent).

A: Approximately 400 large fires occurred across our study extent in 2015/16, and we did intend to sample all of these fires (but from at most one fire) for testing purposes. Please see clarifications on pg 8, lines 7-9.

R: Line 25, Page 8, How do you suspect the model handles non-stationarity? It was built with historical conditions.

A: We meant to imply that the models themselves can be easily updated with new historical data, without having to change the input variables. Underlying relationships may be changing– for instance precipitation of the wettest month or the average early May EVI values – but the models would simply need to be re-trained on updated datasets to integrate such nonstationarities. We have clarified this point on Pg 10, lines 21-25.

R: Table 1: Missing ERC from section 4. Table 1 figure is a bit awkward and might not be needed.

A: We have added ERC to Table 1 and also removed the figure.

R: Figure 1 caption: Use SI units. Figure 1 itself shows MTBS. Why introduce MTBS here? Also, do you require a different training seed within each large fire? Note that these really won't be independent points due to autocorrelation.

A: We've switched the units in Figure 1 to SI. We believe that with revisions to the manuscript, it is clearer why we introduced MTBS here, and also how the different training seeds were used to build 10 different models.

R: Figure 3, I think it is best to just show a single map, but specify the exact date, and perhaps show any large fires that actually occurred.

A: Agreed - we have created a new Figure 3.

R: Figure 5, This is a poor figure. I don't understand what the 10 different points refer to, why the 0.45 cutoff is shown, and didn't realize that there was a model for small fires in addition to large fires. It also looks like this includes data through 2017. If so, please update the caption.

A: We have removed this figure from the manuscript.

Please also note the supplement to this comment:
https://www.earth-syst-sci-data-discuss.net/essd-2017-136/essd-2017-136-AC1-supplement.pdf
* * *
[Figure]

[Figure]

Training Seeds

MTBS >1000ac Fire Perimeters

MODIS >1000ac Burned Areas

**Fig. 1.** Example of how the MODerate-resolution Imaging Spectroradiometer (MODIS) Burned Area (BA) dataset was used to draw 10 random sample seeds from within large fires. Each seed, across all large fires in

[Figure]

**Fig. 2.** Receiver Operating Curve (ROC) for an independent testing dataset of small and large fires that occurred from 2015-2016. Sensitivity and (1-Specificity) values are shown for the point where large fire
Interactive
comment

**Prediction for July 30, 2015**

**Large Fire Probability (x100)**
High : 94

Low : 3

☐ August 2015 MTBS Fire Perimeters
☐ EPA Ecoregions

0   80   160 Mi
0   140   280 Km

**Fig. 3.** Large fire probability for the week of July 30, 2015. MTBS fires greater than 405 hectares, and that started in August 2015, are overlaid on the map.

[Figure]

**Fig. 4.** False positive (FP) and false negative (FN) rates of an independent testing dataset of small and large fires from 2015-2016, mapped across EPA level three ecoregions. No testing data was available for

**Supplement:**

**A weekly, near real-time dataset of the probability of large wildfire across western US forests and woodlands**

Miranda E. Gray[1], Luke J. Zachmann[1,2], Brett G. Dickson[1,2]

[1] Conservation Science Partners, Inc., Truckee, CA 96161, USA
[2] Lab of Landscape Ecology and Conservation Biology, Northern Arizona University, Flagstaff, AZ, 86011, USA

5  *Correspondence to*: Miranda E. Gray (miranda@csp-inc.org)

**Abstract.** There is broad consensus that wildfire activity is likely to increase in western US forests and woodlands over the next century. Therefore, spatial predictions of the potential for large wildfires have immediate and growing relevance to near- and long-term research, planning, and management objectives. Fuels, climate, weather, and the landscape all exert controls on wildfire occurrence and spread, but the dynamics of these controls vary from daily to decadal timescales.

10  Accurate spatial predictions of large wildfires should therefore strive to integrate across these variables and timescales. Here, we describe a high spatial resolution dataset (250-m pixel) of the probability of large wildfire (>405 ha) across all western US forests and woodlands, from 2005 to the present. The dataset is automatically updated on a weekly basis and in near real-time (i.e., a one-week lag) using Google Earth Engine and a 'Continuous Integration' pipeline. Each image in the dataset is the output of a machine-learning algorithm, trained on 10 independent, random samples of historic small and large wildfires,

15  and represents the predicted probability of an individual pixel burning in a large fire. This novel workflow is able to integrate the short-term dynamics of fuels and weather into weekly predictions, while also integrating longer-term dynamics of fuels, climate, and the landscape. As a near real-time product, the dataset can provide operational fire managers with immediate, on-the-ground information to closely monitor changing potential for large wildfire occurrence and spread. It can also serve as a foundational dataset for longer-term planning and research, such as strategic targeting of fuels management, fire-smart

20  development at the wildland urban interface, and analysis of trends in wildfire potential over time. Weekly large fire probability GeoTiff products from 2005 through 2017 are archived on Figshare online digital repository with the DOI 10.6084/m9.figshare.5765967 (available at https://doi.org/10.6084/m9.figshare.5765967.v1). Near real-time weekly GeoTiff products and the entire dataset from 2005 on are also continuously uploaded to a Google Cloud Storage bucket at https://console.cloud.google.com/storage/wffr-preds/V1, and available free of charge with a Google account. Near real-time

25  products and the long-term archive are also available to registered Google Earth Engine (GEE) users as public GEE assets, and can be accessed with the image collection ID 'users/mgray/wffr-preds' within GEE.

**1 Introduction**

Operational versus research-driven needs for wildfire prediction operate at different spatiotemporal scales, aiming either to understand the risk posed by individual fires or over the course of a fire season, or to understand the broad-scale

characteristics of fire regimes. For example, operational needs emphasize the short-term controls on fire (e.g., occurring over days to months; Brillinger et al., 2003; Martell et al., 1989; Sullivan, 2009a, 2009b) and largely ignore the longer term controls (e.g., occurring over years to decades). By contrast, research to predict across longer time frames and often larger spatial scales will omit the real-time weather patterns that drive fire occurrence (Krawchuk and Moritz, 2014; Littell et al.,

5    2009; Urbieta et al., 2015). While many models and datasets exist to support these needs, they also reflect these different and non-overlapping scales. We sought to fill this gap by developing a dataset of large wildfire probability that integrates across spatiotemporal scales in an empirical framework, at a high spatial resolution (250-m pixel) and moderate temporal resolution (weekly), and across the western US. The resulting dataset is intended to meet multiple objectives of local to national research, management, and planning efforts.

10    A well-developed approach to incorporate the dynamic short-term drivers of wildfire is to simulate the spread of a fire with physics-based models (Finney, 2004; Sullivan, 2009c; Tymstra et al., 2010). Tools that allow for these simulations, such as Farsite (Finney, 2004), Flammap (Finney, 2006), and FSPro (USDA Forest Service), are used widely during wildfire incidents and in real time to understand the potential spread and behavior of burning fires. These tools can provide critical information for individual or localized fire probability and behavior in real time, but are limited in their ability to elucidate

15    regional and cross-regional fire risk at similar time frames, and are dependent on fuels data, e.g. from Landfire (Rollins, 2009), that are not updated in real time and often not for years at a time. Although the work described herein does not attempt to model the risk or behavior posed by individual fires, it is meant to provide high-resolution data of fire probability in near real time and also across regional extents, drawing on continuously updated fuel and weather predictors. Therefore, it provides a needed, complementary dataset to existing tools that operate on short timescales.

20    By simulating individual fires across time and space, physics-based models can also be scaled-up to predict the long-term potential of fire at every point on a landscape (Finney et al., 2011; Parisien et al., 2005). This approach is commonly used for longer-term planning of fuel treatments and other fire risk planning and assessments (Haas et al., 2013; Thompson et al., 2017). Because they require detailed specifications of many model inputs and are sensitive to assumed model parameterizations, these landscape-scale simulations can be input- and computationally-intensive (Parisien et al.,

25    2012a; Varner et al., 2009). At regional to national scales, user-intensive demands may also constrain the ability of analysts and planners to update datasets at both broad spatial scales and decision-relevant timescales. For example, predictive datasets may need to be updated according to changes in fuel that occur within a fire season and on an inter-annual basis.

Alternative methods to predict fire occurrence relate empirical fire data to environmental predictors in statistical models (Gray et al., 2014; Preisler et al., 2016; Stavros et al., 2014). Data availability in this case, namely the spatio-

30    temporal alignment of accurate and high-resolution fire, weather, and fuels data, also acts as a constraint on either the spatial or temporal scale of analysis (Taylor et al., 2013). However, such statistical methods are common in predicting fire occurrence on a macro-scale because they can draw on coarse scale data to overcome this constraint (Krawchuk et al., 2009; Moritz et al., 2012; Parisien et al., 2012b). Owing to the flexibility of model specification and data inputs, as well as

increasingly accurate and high-resolution observational data, statistically-based empirical models can integrate both the dynamic short and long-term controls on fire potential.

Indeed, recent studies have explicitly compared the role of temporal scale in predicting fire occurrence, and have shown that long-term normals and variability in climate and vegetation provide complementary predictive power (Abatzoglou and Kolden, 2011, 2013; Parisien et al., 2014; Riley et al., 2013). For example, long-term climate exerts an influence on the flammability (e.g., due to biomass production, vegetation composition, and average fuel moisture) of a fuel bed, but weekly and sub-weekly weather will moderate fuel moisture in a site-specific way. Similarly, short-term disturbance events such as previous burns can regulate biomass production and subsequent fire risk on inter-annual timescales (Parisien et al., 2014; Parks et al., 2015). It follows that predictive datasets of wildfire potential should strive to integrate across complex, dynamic interactions at short- and long-term timescales. Here, we describe a time-series of the probability of large fire, updated on a weekly basis and in near real-time (i.e., to the present week) to integrate the weekly, short-term controls on fire occurrence, but which also considers the longer-term influences of land use, disturbance, climate, and topography. The complete dataset (2005-present) can also be considered a foundational dataset for understanding long-term, probabilistic exposure of forests and woodlands to large fires.

**2 Methods**

**2.1 Modelling**

We modeled the probability of large fire occurrence, which we defined as the probability that an area on the landscape will burn in a large (i.e., > 405 ha) fire, conditional on either an ignition event or fire spreading to that area. While defining large fire size is somewhat arbitrary, 405 ha is commonly used to distinguish large from small fires in western US forests (e.g., Westerling, 2006), and fires > 405 ha accounted for approximately 95% of area burned in western forests and woodlands from 1992-2015 (Short, 2017). Additionally, our method only focused on the probability of a large fire irrespective of ignition likelihood or sources. Ignitions are non-random events that adhere to spatial patterns of anthropogenic or lightning activity (Balch et al., 2017), which are not accounted for in this dataset.

We used a random forest (RF) classification algorithm (Breiman, 2001) to train predictive models of large fire probability. RF is a machine learning technique that recursively partitions variables to classify an outcome of interest, in this case small or large fire events. Multiple classification trees are fit to bootstrapped samples of the training data, but at each node, only a fraction of randomly selected predictors are available for the binary partitioning. The randomized process of recursive partitioning uncovers hidden structures in the data without over-fitting, and yields strong predictive models (Prasad et al., 2006). Thus, RF is an ideal method to predict fire occurrence across broad and diverse ecoregions, where high dimensionality is needed to account for unforeseen interactions between climate, fuels, and the landscape (Cutler et al., 2007).

The binary response variable in our RF models was a point on the landscape where there was an ignition event that resulted in a small fire (i.e., < 405 ha; '0' response) or that historically burned in a large fire (i.e., > 405 ha; '1' response). Therefore, model outputs (i.e., raster maps) can be interpreted as reflecting the probability that a given area on the landscape will burn in a large fire, conditional on either an ignition or spread of fire to that area. We sampled large fire points from the
5    MODerate-resolution Imaging Spectroradiometer (MODIS) Burned Area (BA) dataset (Roy et al., 2008), which is a 500-m remote sensing product that contains the day-of-burn, and we sampled small fire points from a database of reported fires in the United States (Short, 2014, 2017) that contains the day-of-ignition (Sect. 2.2). To avoid autocorrelation within large fires, we drew the large fire samples from at most one large fire (see Sect. 2.2), which we matched with an equal sized random sample of small fires, to build a single RF model across the entire western US. While autocorrelation is invariably present
10    within individual fires, burning conditions can also be quite heterogeneous over the course of a single large fire (Turner, 2010). We therefore repeated the above sampling and model building protocol with 10 different random seeds, such that each of 10 RF models were not entirely independent but contributed slightly novel information to a mean prediction across those 10 models. This type of ensemble modeling provides a means of producing models that are more accurate than the individual models that make them up, and depicting the variance across predictions, which is critical for risk assessment
15    (Dietterich, 2000; Palmer et al., 2005).

Using 10 trained RF models, we created weekly spatial predictions of the mean and standard deviation of large fire probability at 250-m resolution across western US forests and woodlands. Spatial predictions were created for every week from 2005 through the present. See Sect. 4 below that describes the process by which new predictor data acquisitions are automatically and continuously integrated into near real-time predictions and uploaded to the cloud. Models were trained and
20    spatial predictions created within Google Earth Engine (GEE; Gorelick et al., 2016), which is a cloud-based platform that makes terabyte-scale analysis available on an extensive catalog of satellite imagery and geospatial datasets.

**2.2 Response Variables**

We sampled large fires by retaining MODIS pixels that were within eight days of reported burn date of neighboring burned pixels. This boosted our confidence that connected pixels were part of the same fire (Archibald and Roy, 2009),
25    which we also required to be connected to ≥ 15 other pixels (≅405 ha). We then used the MTBS dataset to delineate the perimeters of annual large wildfires (excluding prescribed fires), and sampled daily MODIS burned area pixels in a given year from within these perimeters. We masked burned areas according to forest or woodland land cover types classified in the 2001 US National Land Cover Dataset (NLCD, 30-m resolution; Homer et al., 2007) before drawing 10 random sample seeds across all large fires ($n \cong 900$ in each seed) from 2005-2014. Each individual large fire sample was taken as the
30    centroid of a 500-m pixel (Figure 1). We used the 2001 NLCD product because it represents the closest complete land cover prior to the fires selected for training data in this analysis.

We drew random samples of small wildfires (also excluding prescribed fires) from the US Fire Occurrence Dataset (FOD; Short, 2014, 2017), masked by NLCD forest and woodland cover. We did not draw small samples from the BA dataset because the estimated minimum detectable burn size is approximately 120 ha, which means that smaller fires are grossly underestimated (Giglio et al., 2009; Roy and Boschetti, 2009). Within each Environmental Protection Agency (EPA) level III ecoregion in the western US, we paired an equal sized random sample of small fires with each of the 10 large fire sample seeds, resulting in spatially balanced, 1:1 training datasets across diverse ecoregions.

**2.3 Predictor Variables**

We derived predictor variables that describe the land surface and climate over multi-year, long-term time frames. Similarly, we derived predictor variables that describe the land surface and weather over weekly, short-term time frames (Table 1). Specifically, an individual large or small fire sample was spatially related to long-term predictors derived over a multi-year period and short-term predictors derived over the week before and after fire occurrence. The integration of predictors in this way resolves the dynamic probability of large fire into long-term drivers of fire, and short-term land surface and ambient conditions directly leading up to and following a fire event. To account for the difference in spatial scales between a large fire and the native resolution of spatial predictors (i.e., ranging from 250 m to 4 km), we used a moving window to derive the mean value of predictors within a circular kernel with a radius of 1135 meters. Predictor variables that were not in a native 250-m resolution were resampled using bilinear interpolation.

**2.3.1 Long-term Land Surface Variables**

To characterize long-term biomass production and water content per pixel, respectively, we used the Enhanced Vegetation Index (EVI, 250-m resolution) from the MODIS MOD13Q1 v006 product (Didan, 2015) and the Normalized Difference Water Index (NDWI, 500-m resolution), derived from the MODIS MCD43A4 v006 product (Schaaf, 2015). MODIS EVI and the Normalized Difference Vegetation Index (NDVI) both provide proxies for total vegetation, but EVI is more sensitive to canopy variations in densely vegetated areas (Huete et al., 2002). We used a multi-year time-series of EVI to capture the variability in overall biomass production across the western US, but also as a basis to capture variability in sub-pixel vegetation dynamics (e.g., Helman et al., 2015). We also included EVI to capture longer-term changes in fuel abundance due to prior burns, based on findings that forested ecoregions have shown large to moderate post-fire reductions in MODIS NDVI over a ten year period (Yang et al., 2017).

The NDWI was originally proposed as a complementary vegetation index to NDVI and EVI to detect vegetation liquid water content (Gao, 1996), and has since been shown to relate strongly to the total water content per pixel (Cheng et al., 2006; Maki et al., 2004). Similar to EVI, we included a multi-year time-series of NDWI to capture moisture gradients across space. NDWI has also been successful in estimating vegetation moisture and fire hazard when coupled with an estimate of the total vegetation, and so the interaction between EVI and NDWI may provide important information about pixel-wise fuel moistures (Maki et al., 2004).

Each of the NDWI and EVI products used in our analysis were 16-day composites computed from atmospherically corrected, bi-directional daily surface reflectance. MOD13Q1 contains pixel quality information and MCD43A4 contains pixel and band quality information. For both products, we only retained pixels that were free of ice and snow. We extracted five percentile values (10, 25, 50, 75 and 90%) of EVI and NDWI from 2000 (the year MODIS was deployed) to the approximate date of each fire occurrence. These values provided at least five complete years of observed EVI and NDWI prior to the occurrence of a given fire.

To characterize the land surface as modified by humans over the long-term, we included indices of human modification for the years 2001 and 2011 (Conservation Science Partners Inc., 2016; 30-m resolution). The index quantifies the cumulative degree of modification of natural lands attributable directly to energy, residential and commercial, transportation, and agricultural development. We hypothesized that more developed landscapes, because they are less natural and generally more fragmented, are less likely to burn in large fires. We also used the associated residential and commercial development dataset (Conservation Science Partners Inc., 2016; 30-m resolution) to compute the Euclidean distance to urban development in 2001 and 2011. Urban development in this case was approximated by a 'moderate' value of residential and commercial development, which is roughly equivalent to the 'built up moderate' class in the NLCD, except that it removes exaggerated effects of roads. We assumed that suppression resources and mandates are more readily accessed closer to urban centers and thus constrains the likelihood of large fires. Lastly, we used the Shuttle Radar Topography Mission digital elevation data (Farr et al., 2007) to characterize topographic variables, namely, elevation, slope, aspect, and terrain roughness (standard deviation of elevation), each at a 30-m resolution.

**2.3.2 Long-term Climate Variables**

We incorporated predictors computed from monthly climatological normals of temperature and precipitation for the period 1981-2010, as derived from the Parameter-elevation Regressions on Independent Slopes Model (PRISM; 800-m resolution; Daly et al., 1994). We selected five metrics that summarized long-term annual means, extremes, and seasonality of temperature and precipitation, and which have been used previously to capture the amount and dryness of biomass to predict fire occurrence (Krawchuk et al., 2009; Moritz et al., 2012). These metrics included annual precipitation, precipitation of the warmest month, mean temperature of the wettest month, mean temperature of the warmest month, and temperature seasonality (i.e., the standard deviation of mean monthly temperatures; O'Donnell and Ignizio, 2012).

**2.3.3 Short-term Land Surface Variables**

We characterized short-term live vegetation abundance and condition, as well as pixel water content, with the single EVI and NDWI observations in the month prior to fire occurrence. We included the absolute index value, as well as anomalies from the closest day-of-year in years prior and from the five percentile values. These short-term indices are meant to capture the absolute vegetation abundance and condition prior to burning, and also the deviance from its long-term state. Specifically with EVI, short-term deviances have been shown to correlate well with forest fire probability (Bisquert et al.,

2013). NDWI, when coupled with EVI, has also been shown to contribute to fire risk on sub-monthly timescales (Maki et al., 2004).

We used the MODIS MOD11A2 daytime Land Surface Temperature (LST) eight-day composites (1-km resolution; NASA LP DAAC, 2015), which represent average values of clear-sky LSTs, to similarly characterize the ground temperature immediately leading up to a fire occurrence. Due to a feedback between LST and near-surface humidity, remotely sensed LST has been used to predict vapour pressure deficit, which itself is a good short-term predictor of fine dead fuel moisture and fire danger (Boer et al., 2017; Nolan et al., 2016). We included both the absolute value of LST from the eight days prior to fire, as well as the LST anomalies from the five percentile values and from the closest day-of-year in years prior.

**2.3.4 Short-term Weather Variables**

Standard meteorological variables known to influence the daily fire and fuel environment were taken from the GRIDMET gridded daily surface meteorological dataset (4-km resolution; Abatzoglou, 2013). We incorporated the total precipitation, mean minimum and maximum temperature, mean minimum and maximum relative humidity, mean wind speed and direction and the mean Palmer drought severity index (PDSI) for the two weeks surrounding fire occurrence.

Standard weather variables have also been compiled into indices that more directly address the processes by which they effect fires and fuels, including the Energy Release Component (ERC), the Burning Index (BI), and 100- and 1000-hr dead fuel moistures (fm100 and fm1000). These indices are components of the US National Fire Danger Rating System (NFDRS) and are derived from models built on the combustion physics and moisture dynamics of the fuel environment, here assuming a consistent fuel model 'G' typified by short needle pine and heavy dead loads (Abatzoglou, 2013; Schlobohm and Brain, 2002). The fm100 and fm1000 represent the modeled moisture content of large dead fuels and are functions of the latitude, day-of-year, temperature, relative humidity, and precipitation duration over the previous 24 hours (fm100) or seven days (fm1000). ERC is a cumulative fuel moisture index reflecting the contribution of all live and dead fuel moistures on the potential heat release, and is also an input into the BI, which additionally incorporates the potential rate of fire spread. GRIDMET assumes that the persistent fuel environment includes all size classes of dead fuels, as well as herbaceous and woody live fuels, and all contribute to the derived values of these indices. We incorporated the mean values of ERC, BI, fm100, and fm1000 in the two weeks surrounding fire occurrence.

**3 Dataset Evaluation**

Using all training data from 2005-2014 (i.e., no independent testing data), we compared models and extracted variable importance in R using the 'caret' package (Kuhn, 2008). Variable importance was determined using a normalized mean decrease in accuracy when that variable was not included. Across the 10 models, overall accuracy was consistently

between 0.7 and 0.72 and area under the receiver operating curve (AUC) was consistently from 0.78-0.79. More importantly, we examined the main differences across models in the top 20 important variables. Out of 76 total predictor variables, human modification or distance to urban development, the mean BI, and the lowest percentiles of NDWI were consistently in the top five variables. Topographic roughness and mean ERC were consistently in the top 10 variables. Slope, LST, mean 1000-hr FM, the highest percentiles of NDWI, precipitation of the warmest month, and EVI (both short- and long-term) were consistently in the top 20 variables. Other variables showed up inconsistently in the top 20 variables.

To independently evaluate the model on data from 2015-2016, we used the MODIS BA and FOD datasets to draw a testing sample from within all large fires, and an equal-sized random sample of small fires (response value of '0' and '1', respectively; $n \cong 400$ large fires). Again, large samples were taken as the centroid of 500-m pixels. Using weekly predictions (i.e., raster maps) of large fire probability in 2015 and 2016, we extracted predicted values at the time (i.e., the closest prediction in time prior to fire occurrence) and location of individual testing points. We used the R package 'OptimalCutpoints' (López-Ratón et al., 2014) to determine an optimal cutoff between zero and one that simultaneously maximized the sensitivity (true positive rate) and specificity (true negative rate) of predictions. In this case, using a probability cutoff of 0.45 to predict binary large (> 0.45) versus small (< 0.45) fire resulted in the greatest rate of true positives and negatives in our testing datasets. Based on an optimal cutoff of 0.45 and two years of independent data, the overall accuracy of the dataset was 0.79, and the area under the receiver operating curve (ROC) curve was 0.88 (Figure 2). We took another step to visualize model performance by mapping the rate of false positives and false negatives (i.e., the number of false positives or false negatives normalized by the number of testing samples) within each EPA level III ecoregion (Figure 3).

**4 Continuous Integration**

We developed a continuous integration (CI) 'pipeline' to generate new predictions as soon as the dynamic predictors upon which the model is conditioned become available in GEE. The refresh rate of each predictor varies based on the data sources. For example, gridMET assets are updated approximately every two days, whereas the MODIS products are updated approximately every eight days (Table 1). The pipeline, which tests for the availability of predictors against the requirements of the model, runs on a schedule — compiling each morning at 4am Pacific Standard Time. If all of the criteria are met, a new prediction is generated and appended to the existing collection. We used GitLab.com because GitLab offers continuous integration (CI) services at no cost. The builds are executed using a custom Docker image, which is a bare-bones Ubuntu image configured with the Google Earth Engine Python API client library and its dependencies.

**5 Band Descriptions**

Band 1 - 'mean' :  Mean probability of large fire across 10 trained models. Values range from 0-1.
Band 2 - 'stdDev' : Standard deviation of the probability of large fire across 10 trained models.

Band 3 - 'modis_QA' : For the MODIS products described above, only good quality pixels were retained for model training, but all pixels were retained when creating spatial predictions. Therefore, final predictions contain a band named 'modis_QA' which indicates if one of the short-term MODIS predictors (i.e., MOD13Q1, MCD43A4, or MOD11A2 immediately preceding the prediction date) had unreliable quality.

5       0 = All MODIS pixels were processed and good quality

        1 = At least one MODIS pixel was not processed or had bad quality

**6 Data Availability and Source Code**

        Weekly large fire probability GeoTiff products from 2005 – 2017 are archived on Figshare online digital repository with the DOI 10.6084/m9.figshare.5765967 (available at https://doi.org/10.6084/m9.figshare.5765967.v1). Near real-time
10    weekly GeoTiff products and the entire dataset from 2005 on are also continuously uploaded to a Google Cloud Storage bucket at https://console.cloud.google.com/storage/wffr-preds/V1, and also available free of charge with a Google account. Near real-time products and the long-term archive are also available to registered GEE users as public GEE assets, and can be accessed with the Image Collection ID 'users/mgray/wffr-preds' within GEE. All source code is available at a GitLab repository (https://gitlab.com/wffr).

15  **7 Conclusions**

        The dataset we describe here of weekly predictions of the probability of large forest or woodland fire across the western US invokes interacting effects over multiple timescales that contribute to a site's dynamic fire potential. By drawing on weather, climate, and land surface dynamics at multiple timescales to predict large wildfire probability at a high spatial and temporal resolution, this dataset fills a gap in existing datasets. The result is highly relevant to research, planning, and
20    management objectives that span the western US, ranging from short-term outlooks to long-term planning.

        More strategic planning for fuels management is critically needed to adapt to an inevitable increase in wildfire in the west in the coming decades (Schoennagel et al., 2017). For instance, fuels treatments as currently implemented are limited in their ability to mitigate broad scale effects of wildfire, because it's relatively rare that treatments actually encounter wildfire (Barnett et al., 2016). Strategically targeting areas for treatment based on large wildfire potential, coupled
25    with estimates of burn severity, will lead to more cost- and ecologically-effective decisions (Scott et al., 2016; Thompson et al., 2017). However, tools currently used for this purpose are often built off of input- and computationally-intensive stochastic simulation models that may constrain the ability to update results at both broad spatial scales and timescales concurrent with the changing fire environment. For example, the Wildland Fire Potential dataset is available for the entire US at 270-m resolution, and describes fire potential as of 2007, 2012, and 2014 (Dillon et al., 2015). The dataset we describe
30    here is automatically updated to match the dynamics of the fuel and fire environment, which can easily change and critically effect fuels management decisions on annual timescales.

Another area where probabilistic fire exposure analysis can help with strategic fuels and fire planning is at the wildland urban interface (WUI; e.g., Haas et al., 2013). WUI lands in the western US have expanded dramatically over the past few decades, and roughly 40% of these lands are predicted to experience moderate to large increases in the probability of wildfire in the next 20 years (Schoennagel et al., 2017). Considering also that a large percentage of potential WUI lands are still undeveloped, strategic planning for both fuels management and infrastructure development can make communities more resilient to wildfire. This dataset can help guide development plans at multiple scales (e.g., city, county, or state), drawing on a rich time series that gives analysts and planners access to the observed trends, means, and extremes of the potential for large wildfire over time.

In contrast to longer-term predictions, near real-time predictions of large fire potential provide operational fire managers with immediate, on the ground information to closely monitor how changing conditions affect active fires, and the likelihood that fire suppression will require outside resources. In the US, near real-time predictions are widely used during the peak fire season (Owen et al., 2012). Available products through the US Predictive Services program (http://psgeodata.fs.fed.us/) and the Wildland Fire Assessment System (www.wfas.net; Preisler et al., 2016), consider fuel and weather conditions changing on daily to weekly timescales, while ignoring longer-term climate and fuel variability that moderate a site's current fire potential. Simulation models that are used in near real-time, such as FARSITE, provide critical information for individual or localized fire probability and behavior, but are limited in their ability to elucidate real-time regional and cross-regional fire risk, and are additionally dependent on fuels data, e.g. from Landfire (Rollins, 2009), that are not updated in real time. The dataset described here provides near real-time predictions across the western US, while simultaneously accounting for dynamic fuel and landscape compositions that are shaped over the short and long term, and thus is needed addition to operational products of near real-time fire potential.

As the observational record grows longer to include more temporal variability and new normals, we will continue to re-train models on the same basis of predictors and update and evaluate this dataset. This will allow for any non-stationary relationships between wildfire, climate, fuels, and the landscape to be easily integrated into predictions. For example, if underlying relationships such as the precipitation of the wettest month or average early May EVI change in the future, models would simply need to be re-trained on updated datasets to integrate such non-stationarities. In future development, forecasted climate, weather, and fuels data may also be integrated into the analysis in order to create predictions of large fire probability into the future.

**8 Author Contributions**

MEG developed the fire models and data products, with critical contributions from both LJZ and BGD. LJZ developed the continuous integration algorithm. All authors contributed to the paper

**9 Acknowledgements**

This research was supported by the Wilburforce Foundation.

25

30

Table 1. Spatially explicit climate and land surface predictors of large fire probability, including the data source, spatial resolution, and description of how variables were derived from the source data. Gray shading in the table indicates grouping of predictor variables depending on whether they are derived over a short term (sub-monthly) or a long term (multi-year). The accompanying graphic indicates the approximate time period that these variables are drawn from, relative to fire occurrence.

| Predictor | Source | Resolution | Description |
|---|---|---|---|
| *1. Long-term climate variables* | | | |
| Annual precipitation, Temperature seasonality, Precipitation of the warmest month, Mean temperature of the wettest month, Mean temperature of the warmest month | PRISM | 800 m | Derived from 1981-2010 monthly normals. |
| *2. Long-term land surface variables* | | | |
| EVI percentiles | MODIS | 250 m | $10^{th}$, $25^{th}$, $50^{th}$, $75^{th}$, and $90^{th}$ percentiles from 2000 to the date of fire occurrence. |
| NDWI percentiles | MODIS | 500 m | |
| Human modification, Distance to urban development | CSP 2016 | 30 m | Index or distance value at 2001 for fires pre 2011, and 2011 for fires post 2011. |
| Elevation, Slope, Aspect, Topographic Roughness | USGS | 30 m | |
| *3. Short-term land surface variables* | | | |
| EVI absolute and anomalies | MODIS | 250 m | Absolute value and percent change from 2000 to the fire year, immediately preceding fire occurrence. |
| NDWI absolute and anomalies | MODIS | 500 m | |
| LST absolute and anomaly | MODIS | 1 km | |
| *4. Short-term weather variables* | | | |
| 100-hr fuel moisture, 1000-hr fuel moisture, Burning index, Energy Release Component, Precipitation, Temperature, Relative humidity, Specific humidity, Potential evapotranspiration, Solar radiation, Wind speed, Wind direction, PDSI | GridMET | 4 km | Mean values in the two weeks surrounding fire occurrence |

**11 Figure Captions**

Figure 1. Example of how the MODerate-resolution Imaging Spectroradiometer (MODIS) Burned Area (BA) dataset was used to draw 10 random sample seeds from within large fires. Each seed, across all large fires in 2005-2014, was used to train a random forest model to predict large fire probability. MTBS fire perimeters greater than 405 hectares are included because they were used to restrict BA sampling within individual wildfires.

Figure 2. Receiver Operating Curve (ROC) for an independent testing dataset of small and large fires that occurred from 2015-2016. Sensitivity and (1-Specificity) values are shown for the point where large fire probability values >0.45 are classified as a large fire, and values <0.45 are classified as a small fire, since this value was found to simultaneously maximize sensitivity and specificity.

Figure 3. Large fire probability for the week of July 30, 2015. MTBS fires greater than 405 hectares, and that started in August 2015, are overlaid on the map.

Figure 4. False positive (FP) and false negative (FN) rates of an independent testing dataset of small and large fires from 2015-2016, mapped across EPA level three ecoregions. No testing data was available for those ecoregions that are not displayed.

Figure 1

[Figure]

Training Seeds

MTBS >1000ac Fire Perimeters
MODIS >1000ac Burned Areas

Figure 2

[Figure]

Figure 3

[Figure]

Figure 4

[Figure]

---

## Author Comment (AC2) · 12 Apr 2018

**R = Reviewer Comment, A = Author Comment**

R: While this manuscript presents a concept that would be valuable (near real-time large fire probability), I believe it oversells its novelty over existing products and I am suspicious that there are methodological flaws (but there is not enough information in the methods section to tell for certain). As the topical editor has already noted, the products are not available at either Figshare or Google Cloud Storage (I did not try the Google Earth Engine).

A: We've updated the access control list on all assets to public READ (anyone on the internet has read access to the objects). Products are now available at all these locations, and we apologize for the links not working initially.

R: While the authors did locate other near real-time products that deliver similar information (e.g. Preisler et al 2016), they missed other datasets currently commonly used for planning fuel treatments and other management activity (notably Short et al's "Spatial dataset of probabilistic wildfire risk components for the conterminous United States" at https://www.fs.usda.gov/rds/archive/Product/RDS-2016-0034/, which was calibrated to previous fires using Short's Fire Occurrence Database which also provides part of the underpinnings of this current effort). This dataset and similar efforts using the FSim model (Finney et al 2011) already deliver many of the functions that the current effort proposes it is uniquely positioned to fill, including acting as "a foundational dataset for longer-term planning and research, such as strategic targeting of fuels management" (e.g. Scott et al 2016, Thompson et al 2017), "fire-smart development at the wildland urban interface" (e.g Haas et al 2013), and "analysis of trends in wildfire potential over time" (Finney et al 2011, then Short et al's updated dataset, and also the fire potential datasets the authors reference by Dillon). Given the mature state of this previous work and its prevalence, its inclusion in the current manuscript would seem important.

A: We've inserted these important references throughout the manuscript and where appropriate, but also attempted to make it more clear that our effort is not unique in filling these positions, as we had previously implied, but still offers complementary advantages over current efforts. For example, this dataset offers an advantage of being able to more immediately draw on updated data, while maintaining the ability to look at probabilistic exposure across large extents and a high resolution. Specifically, please see pg 2, lines 10-19 and pg 10, lines 15-20.

R: In addition, the authors state that other existing models do not account for long-term fuel and climate variability, but I'm not convinced their model better accounts for these due to a number of reasons: 1) their model uses a number of variables as proxies for
fuel and weather that as far as I know have not been demonstrated in the literature to relate to fuel availability and flammability or to meaningfully measure weather's effect on fire, including PDSI (which in fact has been demonstrated to not be related to large fire activity (Riley et al 2013)), EVI, and NDWI,

A: While it is true that PDSI was not found to be strongly related to lare fire occurrence in Riley et al. 2013, this was independent of ecoregion, bioclimatic zones, vegetation type, and any other predictor variables. There is another study for the western US that found PDSI in the fire season to be correlated to fire activity in specific forested ecoregions (e.g. Abatzoglou and Kolden, 2013). While we did not model large fire probability specific to ecoregions, there are interactions with other predictor variables (e.g. bioclimatic and proxy fuel variables), that might have still made this an important predictor.

A: We were incorrect in stating that NDWI can be used directly for live fuel moisture without also accounting for the amount of vegetation, and also incorrect in stating that EVI was used as a proxy for fuel availability. We've added some text and references to clarify that we are using EVI and NDWI as proxies for long-term biomass production and vegetation dynamics, canopy water content, and live fuel moisture when coupled with EVI. We also clarify that we use these variables on a short-term basis as a proxy for vegetation abundance and condition, and they have been shown to help predict fire. Please see pg 5, lines 18-32 and pg 6, lines 28-32.

R: as far as I can tell, their model has a static development layer (the CSP from 2001 and 2011), thus not capturing development trends except coarsely

A: Yes, this is the most comprehensive dataset of anthropogenic development (including urban development, transportation, energy, and agriculture) covering our study extent, and is unfortunately not yet available at more frequent intervals. While it would be ideal to have an annual development layer, other proxies that are directly attributable to human development and available at an annual timescale, such as nightlights from
the VIIRS satellite sensor, would not be as comprehensive and reliable of a predictor.

R: it's unclear how their model captures prior burns, which they state are important factors in fire spread (perhaps they mean the EVI to do this, and indeed it might, but citations or analysis are needed to show that the EVI captures differences in the preand post-fire landscapes).

A: We did intend for the EVI to capture this, and have added a citation and text on Pg 5, lines 19-21.

R: Other functions that they state their model can accomplish are already delivered during wildfire incidents by a suite of models in the Wildland Fire Decision Support System, including FlamMap, FARSITE, and FSPro – which do in fact function in real time, with runs taking on the order of 15 minutes to 1.5 hours and being delivered the same day to suppression forces.

A: We agree that these functions are already provided in these other tools. We have clarified that the functions and advantages of our current effort can be both different and complementary to the functions of the WFDSS tools. For example, this data is not meant to be used on individual fires or at a local scale, but can still provide near real time data at a high resolution and broad spatial extents. Please see pg 2, lines 11-19.

R: However, the approach taken here is novel and could be a valuable addition to the suite of existing models – if methodological questions can be addressed.

Moving on to model structure, I can't discern from the manuscript what the response variable in random forests is. Is it probability of large fire? Probability of small fire? Wouldn't these two be related and increase together? Is probability that the pixel doesn't burn calculated? Or is it simply predicting binary large versus small fire? Or to predict binary burning by large fire versus not burned?

A: We've clarified the model structure, including the response variable. The response variable was binary large ('1') vs. small ('0') fire, resulting in a classification of the
probability that an area would burn in a large fire. See pg 4, starting line 1.

R: Also, it's unclear how predictor variables were chosen. Did the authors choose a set of variables based on their hunches about what's important? Or was variable importance in random forests used to guide selection?

A: Variables were chosen based on previous literature and hypotheses of what drives large fire. Random forest is a predictive framework that does well with many, often correlated predictors, and can uncover some complex interactions between predictors that need not be specified a priori (see Cutler et al., 2007). Therefore, variables were also chosen independently and based on the fact that their interactions may help predict large fire (eg, EVI and NDWI).

R: I am concerned about a potential flaw in model design: it appears that the authors model probability of each pixel independently – however, probability of burning is not independent and is affected by contagion from neighboring pixels. Each pixel's burning can't be considered an independent event, since burning is spatially related to its neighbors. To that end, each large fire might properly be regarded as the unit of prediction, not individual pixels. If the authors have accounted for this, methods should be shared in the paper.

A: We have accounted for the non-independence of individual burning pixels by taking the mean values of predictor variables in a circular kernel with a radius of 1135 meters. This results in a window size that is approximately the size of a large fire (i.e., 405 ha), and assumes that this entire area has an influence on whether a pixel can burn (e.g., fire may spread from a distant source). See pg 5, lines 13-16.

R: The manuscript is largely lacking in assessment of goodness-of-fit of the model.

A: Please see the revised section 'Dataset Evaluation', in which we have added more detail about variable importance, and accuracy of models when evaluated against training and independent testing data.
R: Also, is it feasible to predict burn probability without first predicting ignition? Burning is predicated on ignition taking place first, and then on spatial contagion: thus, the burn probability of one pixel wouldn't be independent of the nearby ones. Some discussion of this, demonstrated by out-of-bag error rates and additional goodness-of-fit metrics is needed.

A: We have clarified that we are modeling the probability of an area burning in a large fire, conditional on either an ignition event or fire spreading to that area. This is different from modelling overall burn probability that accounts first for the occurrence of an ignition. We've added a caveat in the text that overall burn probability would be influenced by non-random ignitions, which we have not accounted for in our analysis. See pg 3, lines 17-23.

R: Also, better demarcation of prescribed fires versus wildfires is needed.

A: Large prescribed fires were excluded from the data used in our analyses by only selecting large fire samples from the MODIS BA dataset from within MTBS wildfires. Prescribed fires are not included in the FOD dataset, and thus were not included in the small fires. See pg 4, lines 25-27.

R: Lastly, how were the outputs validated? By comparison with other existing burn probability or fire regime datasets?

A: Outputs were evaluated with the training data and independent testing data (i.e., small and large fires that occurred from 2015-2016.). We did not compare our outputs with existing datasets, but we do believe that future comparisons of this sort will be important.

Specific comments follow:

R: Page 2 Line 4: the sentence that ends "characteristics of fire regimes" would seem to require citations. Please add.

A: We've added citations to pg 2, lines 4-5.
R: Page 2 Line 7: also see Finney et al 2011, Preisler et al 2016, and Preisler et al 2004, for example, who have already produced this type of work.

A: We have revised the first paragraph and removed this original line, but included these references elsewhere and where appropriate. See pg.2 lines 21 and 29.

R: Page 2 Line 14: it seems that Finney et al 2011 (FSim paper below) should also be referenced here. The comment that follows is not relevant to FSim ("it requires detailed specification of many model inputs and is highly sensitive to misspecification of these parameters"), which is calibrated to fire occurrence in Short's FOD.

A: Finney et al. 2011 is now referenced here. We believe that the following comment is relevant to FSim, since it is input-intensive (requiring a fuel model assignment itself with many parameters, fuel moisture estimation, and weather parameters). The underlying simulation models (i.e., FlamMap) are sensitive to all of these parameters.

R: Page 2 Line 16-18: I am confused by this comment, since as noted above, FlamMap runs in seconds, FARSITE in minutes, and FSPro in 15-90 minutes, meaning that they can and are updated subdaily during active fires.

A: This comment was meant more in reference to models such as FSim, but we have also clarified what we see as the limitations of this type of model and more real-time models like FARSITE. See pg 2, lines 11-19.

R: Page 2 Line 19-20: these lines state that models like Preisler et al 2016 are constrained by availability of accurate high-resolution fire, weather and fuels data – however, later in the manuscript the authors correctly acknowledge that the Preisler et al 2016 model was run daily last year with updated weather data, with outputs available on the WFAS website.

A: This was meant to imply that the spatial or temporal scale is constrained in such models (e.g., sacrificing spatial scale for temporal scale, or vice versa), but not necessarily the temporal scale alone. We have clarified on pg 2 lines 30-31.
R: Page 3 Line 24-26: I'm curious how fuel type (grass, brush, timber litter, etc) is accounted for in the model, as fuel type is directly related to fire spread and probability.

A: Fuel type was not directly accounted for in the models. Several of the predictor variables are meant to provide meaningful approximations of water and energy balances that determine vegetation types (i.e., bioclimatic predictors, topographic aspect, elevation, and slope). Long-term remotely sensed indices of EVI and NDWI are meant to provide inter-annual summaries of the amount and dryness of vegetation, which also helps differentiate vegetation and fuel types. See pg 5, lines 18-32.

R: Page 4 Line 6: What are "large fire event days"? I'm thinking you mean an individual burned pixel. Please elaborate in this paragraph on how you decided whether burned pixels were part of the same fire – from Figure 1, it looks like you assigned pixels to MTBS fires.

A: We removed this term and have also clarified how we used MTBS to constrain large fire sampling. See pg 4, lines 25-27.

R: Page 4 Line 23: what is meant by "there are methods that may be adapted to associate active fire information with small fire events"?

A: We determined that this sentence was unnecessary and removed it, since it was only meant to suggest a potential different approach in the future.

R: Page 4 Line 24: Why was it important to have the same number of small and large fires? There are many more small fires than large fires in Short's FOD.

A: This was done to maintain a balance in the sampling ratio so that the models were not biased towards predicting small fires (see Breiman, 2001). Because training RF models takes bootstrapped resamples of the data to build trees, when there is extremely imbalanced data, there is a significant probability that a bootstrap sample contains few or even none of the minority class. This would result in a tree with poor performance for predicting the minority class. There is then the risk of loss of infor-

ESSDD
mation from the many small fires, but we have partially attempted to overcome this by taking 10 random sample seeds.

R: Page 4 Line 25-26: There are other (perhaps more effective) ways to remove prescribed burns from your dataset. For example, fire type is an attribute in MTBS. By using April-October fires only, you'd include most Rx burns in northern states like Montana and Idaho, and exclude large southern California fires like the recent Thomas Fire that often take place in December.

A: See response to comment above and pg 4, lines 25-27 and pg 5, line 1.

R: Page 4 Line 30: I must confess I'm not familiar with the NDWI, but when I Google it, USGS calls it the "Normalized Difference Water Index" rather than wetness index, and describes it as being used to discern water from non-water. Are you talking about the same index? It seems improbable that there are two MODIS NDWI's with different calculations: : :but perhaps that is the case. Please clarify. In either case, I've not seen the NDWI used in any studies relating it to fire occurrence. So it is a good choice here? The relationship of canopy moisture and flammability is quite complex and not well understood (see for example McAllister et al 2012). Despite the lack of study of the NDWI, it could be a good predictor in your model, but not enough information on variable importance is presented in the current version for me to assess.

A: You are correct - Gao (1996) first referred to NDWI as the Normalized Difference Water Index, and we have revised this in the text. Although Gao proposed it for remote sensing of vegetation liquid content, we have included some more recent references that find it to correlate with pixel water content, and live fuel moisture when coupled with EVI. See Pg 5, lines 12-25 and mention of NDWI variable importance on pg 8 line 3.

R: Page 5 Line 4-5: If I am understanding correctly, you only used MODIS values from inception up to the date of the fire to assign percentile values. Why not use the whole record? It seems in your current method, the percentile assignments would be
sensitive to the date of the fire (so if a fire occurs at an index value of 100 in 2005 and another fire at a value of 100 in 2010 in the same pixel, these could be calculated to be different percentile values since the underlying distribution of values would be different).

A: We did not use the whole record because fires affect spectral reflectance of the vegetation and surface. Therefore, post-fire MODIS estimates may not give a reliable and consistent estimate of what is driving the fire of interest.

R: Page 5 Line 6-7: What is the index of human modification supposed to signify with regards to burn probability? Why is it included? What was the variable importance score?

A: We hypothesized that more developed landscapes, because they are less natural and generally more fragmented, were less likely to burn in large fires. We also assumed that suppression resources and mandates were more readily called upon nearer to urban development, and so included an intuitive measure of Euclidean distance to urban development. Human modification or distance to urban variables were consistently in the top 5 important variables across the 10 models.

R: Page 5 Line 20: Can you explain more about why the CV of temperature and precipitation is "seasonality"? I don't follow. Similarly, why are temperature of the wettest and driest months and precipitation of the coldest and wettest months included as predictors? Have these been demonstrated to correlate with fire probability? Do they have high variable importance scores?

A: Temperature seasonality, which we've revised to be the standard deviation of mean monthly temperature, is defined by the amount of temperature variation over a year (see O'Donnell and Ignizio, 2012). Amongst the other bioclimatic predictors you mention, we've now only included the temperature of the wettest month, which is meant to describe the coincident interactions of energy and water balances, in the absence of more direct, long-term water balance metrics. This was demonstrated to correlate with

ESSDD
fire probability - see updated references in the text on pg 6, line 24.

R: Page 5 Line 23: I think you are saying that EVI is related to fuel availability. I think you are working only in forested ecosystems, so most times EVI will be correlated with the canopy rather than the understory. However, surface fire propagates in the understory and crown fires are relatively rare. So is EVI really related to fuel availability? Also, see McAllister et al regarding live fuel moisture and flammability.

A: We have revised this to clarify that we are using long-term EVI to characterize biomass production, but not fuel availability explicitly. Our hypothesis was that a multi-year time-series of EVI may differentiate between levels of biomass production across the western US, but also interannual, pixel-wise dynamics of vegetation that may help predict large fire. Please see pg 5. lines 18-24.

R: Also in this paragraph, if you have only five years of data in some cases and you are calculating anomalies, you would have only 5 observations, right? Again, why not use the full MODIS record (or did you)? Also, are the average LSTs for both night and daytime temperatures?

A: Yes, for the closest day-of-year anomalies, there was only one observation from each year. Please see the above response for why we did not use the full MODIS record. The LSTs were for daytime temperatures only, and we have clarified this in the text.

R: Page 6 Line 5: Why was PDSI included as a variable when it has been demonstrated not to be strongly correlated with large fire activity (e.g. Riley et al 2013)?

A: Please see response to the PDSI comment above.

R: Page 6 Line 8: Please state which NFDRS fuel model the ERC was calculated for. I believe Abatzoglou's product is for fuel model G.

A: Yes, this was fuel model G.

ESSDD
R: Page 6 Line 11: fm1000 represents the previous 42 days (1000 hr/24 = 41.666 days).

A: It seems that fm1000 was only calculated by using the previous 7 days (see Schlobohm and Brain, 2002, pg 22), but please correct us if you know this to be different for the GRIDMET dataset.

R: Page 6 Lines 18-29: Were small fires assigned to a single pixel? Please explain why only one year of fires was used in evaluation (do you expect these relationships to be stationary from year to year when there is so much annual variability in area burned?). For the rest of this paragraph and the following paragraph I'm quite confused. I don't understand what the response variable in the model is (as stated above). Also, can you briefly define sensitivity and specificity? If the response variable is probability, how do you define a false negative and false positive?

A: Yes, small fires were assigned to a single pixel. Please see response to comment above for clarification of the response variable. Sensitivity and specificity have been defined in the text. Although predicted probability was extracted at the testing data points, these were assigned a binary predicted response of '0' or '1' based on the optimal cutoff, thereby allowing us to assign false negatives and false positives.

R: Page 7 Line 10: I would like to see each of the bands illustrated by a figure, otherwise it's quite difficult for a reviewer to visualize and assess the product. How prevalent were pixels with a rating of 1 (at least one MODIS pixel was not processed or had bad quality)?

A: Unfortunately, we did not get to this suggestion but really like the idea, and can create this figure for a revised submission, if deemed appropriate.

R: Page 8 Line 7: again, see other literature including but not limited to Thompson et al 2017 and Scott et al 2016.

A: See pg 9, lines 25-26.
R: Page 8 Line 11: There are other products updated daily that account for changes in weather and fuel moisture, including Preisler et al 2016. Some of the inputs to your model appear to be static, including the CSP (human development layer) and it's not clear how past disturbances (burns) are included in your model. Perhaps the EVI captures previously burned areas, but I know of no study that documents that. Have you assessed how your model works in recently burned vs. burned areas?

A: Please see responses above related to the human development layer, and EVI to capture prior burns. In future analysis with this dataset, we would like to evaluate how it compares in recently burned areas across the west, but we have not done this evaluation yet.

R: Page 8 Line 20: It's not clear how this model would provide better information to managers during active fires than the suite of models in WFDSS (FlamMap, FARSITE, and FSPro), which output information on predicted fire intensity, fire spread, and burn probability in near real-time. Please clarify.

A: Please see response to one of your first comments above, related to this, and pg 10, lines 15-20.

R: Figure 1: This figure nicely illustrates how incomplete MODIS data is!! I've noticed this while daily following nearby fires in my area. MODIS often misses surface fires where the canopy is dense or even crown fires where the smoke plume is dense. Is MODIS then a good basis for predicting burned pixels (especially when it can be difficult to eliminate Rx fire)?

A: We chose to use MODIS because it provides the day-of-burn, which can then be related more precisely to predictor variables. While it is rather incomplete, we've noticed that it still captures pixels within most of the MTBS perimeters (as illustrated in this figure), and provided enough data for our modeling purposes.

R: Figure 3: Please present actual values rather than "high" or "low". I don't feel I can

**ESSDD**
validate the product without them.

A: We have added the actual values to this figure, and per a recommendation from another reviewer, have only showed one prediction date. We've also included MTBS fires that occured in the month immediately after this prediction, as another coarse way to evaluate the product.

R: Figure 4: I'm confused here. Why not present at-pixel values? Are these the sum or average of false positives for an ecoregion? I'm also confused as to what these mean: there is always a probability of fire, so what does it mean to have a false negative or false positive if you are predicting probability? Of course, as I said earlier, I'm confused about what the response variable is, so when I understand that perhaps I won't be confused here.

A: We believe that providing at-pixel values would make this figure too hard to read, if we understand you correctly. In an attempt to make it readable and still informative, what we've presented is the rate (e.g., # of false positives/total number of testing data points) in each ecoregion. Although predicted probability was extracted at the testing data points, they were assigned a binary predicted response of '0' or '1' based on the optimal cutoff, thereby allowing us to assign false negatives and false positives. That said, we are very open to revising this figure if deemed appropriate, perhaps by using at-pixel values.

R: Figure 5: I'm confused here too. Is the white squiggly line the probability of small fire and the black squiggly line the probability of large fire? If so, the y-axis is incorrect. Is the vertical white line the date of a small fire, and the black vertical line the date of a large fire? What does it mean to randomly pair a large and small fire? Should they be related? Why in some cases are the black and white trends similar and in some cases different?

A: This figure caused confusion to multiple reviewers, so we have decided to eliminate it from the manuscript.
References:

Abatzoglou, J. T. and Kolden, C. A.: Relationships between climate and macroscale area burned in the western United States, Int. J. Wildl. Fire, 22(7), 1003–1020, 2013.

Breiman, L.: Random forests, Mach. Learn., 45(1), 5–32, doi:10.1023/A:1010933404324, 2001.

Cutler, D. R., Edwards, T. C., Beard, K. H., Cutler, A., Hess, K. T., Gibson, J. and Lawler, J. J.: Random Forests for Classification in Ecology, Ecology, 88(11), 2783–2792, doi:10.1890/07-0539.1, 2007.

Gao, B. C.: NDWI - A normalized difference water index for remote sensing of vegetation liquid water from space, Remote Sens. Environ., 58(3), 257–266, doi:10.1016/S0034-4257(96)00067-3, 1996.

O'Donnell, M. S. and Ignizio, D. A.: Bioclimatic Predictors for Supporting Ecological Applications in the Conterminous United States, U.S Geol. Surv. Data Ser. 691, 10, 2012.

Schlobohm, P. and Brain, J.: Gaining an understanding of the National Fire Danger Rating System, 2002.

Please also note the supplement to this comment: https://www.earth-syst-sci-data-discuss.net/essd-2017-136/essd-2017-136-AC2supplement.pdf ESSDD